# Non-Ionic Surfactants for Stabilization of Polymeric Nanoparticles for Biomedical Uses

**DOI:** 10.3390/ma14123197

**Published:** 2021-06-10

**Authors:** Hernán Cortés, Héctor Hernández-Parra, Sergio A. Bernal-Chávez, María L. Del Prado-Audelo, Isaac H. Caballero-Florán, Fabiola V. Borbolla-Jiménez, Maykel González-Torres, Jonathan J. Magaña, Gerardo Leyva-Gómez

**Affiliations:** 1Laboratorio de Medicina Genómica, Departamento de Genómica, Instituto Nacional de Rehabilitación Luis Guillermo Ibarra Ibarra, Ciudad de México 14389, Mexico; hcortes@inr.gob.mx (H.C.); fvbj@hotmail.com (F.V.B.-J.); 2Departamento de Farmacología, Centro de Investigación y de Estudios Avanzados del Instituto Politécnico Nacional, Ciudad de México 07360, Mexico; hector.hernandez@cinvestav.mx (H.H.-P.); hiram.qfohead@gmail.com (I.H.C.-F.); 3Departamento de Farmacia, Facultad de Química, Universidad Nacional Autónoma de México, Ciudad de México 04510, Mexico; q901108@hotmail.com; 4Escuela de Ingeniería y Ciencias, Departamento de Bioingeniería, Tecnológico de Monterrey Campus Ciudad de México, CDMX, Ciudad de México 14380, Mexico; luisa.delpradoa@gmail.com; 5CONACyT-Laboratorio de Biotecnología, Instituto Nacional de Rehabilitación Luis Guillermo Ibarra Ibarra, Ciudad de México 14389, Mexico; mikegcu@gmail.com

**Keywords:** non-ionic surfactant, nanoparticle, polysorbates, poly(vinyl alcohol), poloxamer, stability, quality by design

## Abstract

Surfactants are essential in the manufacture of polymeric nanoparticles by emulsion formation methods and to preserve the stability of carriers in liquid media. The deposition of non-ionic surfactants at the interface allows a considerable reduction of the globule of the emulsion with high biocompatibility and the possibility of oscillating the final sizes in a wide nanometric range. Therefore, this review presents an analysis of the three principal non-ionic surfactants utilized in the manufacture of polymeric nanoparticles; polysorbates, poly(vinyl alcohol), and poloxamers. We included a section on general properties and uses and a comprehensive compilation of formulations with each principal non-ionic surfactant. Then, we highlight a section on the interaction of non-ionic surfactants with biological barriers to emphasize that the function of surfactants is not limited to stabilizing the dispersion of nanoparticles and has a broad impact on pharmacokinetics. Finally, the last section corresponds to a recommendation in the experimental approach for choosing a surfactant applying the systematic methodology of Quality by Design.

## 1. Introduction

Surface active agents, commonly known as “surfactants”, are molecules that decrease surface and interfacial tension at the interfaces between solids, liquids, and gases, acting as dispersants, wetting agents, emulsifiers, and detergents [1]. Furthermore, surfactants can maintain the stability of the dispersed phases through the primary interaction at the interface, regulating the exchange of energy and matter in natural and synthetic processes. Thus, the participation of surfactants in the interaction of apparently incompatible phases is crucial [2].

A dispersed system consists of one substance distributed (dispersed phase) in discrete units in a second substance (continuous phase). Most of the nanoparticle manufacturing methods in the biomedical field involve forming a liquid/liquid stable dispersed system with the contribution of surfactant agents to produce a new colloidal type solid/liquid dispersed system. The initial globule size of emulsified dispersed systems is greater than the colloidal particle size at the end of the manufacturing process, combined with the presence of a high surface free energy and, therefore, the tendency of resulted nanoparticles (NP) to flocculate and coagulate can be observed [3]. At the same time, the stability of NP as a dispersed system in an aqueous medium is a fundamental challenge and a critical subject of argumentation in most studies [4]. For this reason, the presence of surfactants is essential before, during, and after the formation of the NP [5].

The presence of a surfactant affects the particle size, polydispersity index (PDI), drug loading, zeta potential value, and correlation with apparent physical stability [6]. For this reason, traditionally, the focus of surfactants is restricted to the stability phenomena of NP [7]. However, the biological interaction highly depends on the surface phenomena of the NP and, consequently, on surfactant agents [8]. The arrangement of surfactants at biological interfaces contributes to cell, tissue, and organ homeostasis. Currently, it highlights the trend of surfactant therapies to lessen alterations in surface tension derived from inflammatory processes. However, the levels of industrial surfactants in the environment have always been a matter of concern and monitoring [9,10,11]. Current applications of surfactants in the manufacture of NP for biomedical applications seek a vectorization phenomenon to facilitate drug release at receptor sites [12]. In this regard, the participation of surfactants represents a multifunctional ingredient that usually requires adsorption by covalent crosslinking to guarantee better performance in biological pathways [13,14].

Polymeric NP are carriers that predominate in biomedical applications, while non-ionic surfactants confer high biocompatibility in most methodologies. For several decades, most nanoparticle formulations have included one of the following three excipients as a surfactant: polysorbates (PS), poly(vinyl alcohol) (PVA), or poloxamers. This broad trajectory of study has allowed an abundant exploration of technological benefits and formulation limitations.

This work is a tribute to the principal non-ionic surfactants utilized to manufacture polymeric NP in the biomedical field. It focuses on the three leading excipients for stabilization: PS, PVA, and poloxamers. The purpose of this work is to offer an overview of each type of non-ionic surfactant, including a general description of the types of molecules in the family, general applications, obtention of derivatives in search of novel properties, and formulation of NP with tables that condense aspects of the physicochemical parameters of the NP according to the type of stabilizer. Moreover, we mention a brief section on toxicity aspects, a critical section on advantages and disadvantages, and a section on drawbacks and future. Finally, this review presents an analysis of the influence of surfactants on the interaction with biological barriers and a narrative and comprehensive description of the main variables involved in the methodological selection of surfactants through the Quality by Design strategy (QbD).

## 2. Use of Surfactants for Nanoparticle Stabilization

Surfactants are crucial excipients in the synthesis of NP; they are amphiphilic molecules characterized by a hydrophilic head group (ionic or non-ionic) and a hydrophobic tail (Figure 1a). The amphiphilic nature of surfactants has been exploited to stabilize hydrophobic nanomaterials in aqueous media [15]. Hydrophobic regions interact with NP surfaces, and hydrophilic regions interact with water (Figure 1b), thus providing colloidal stability and improving dispersion stability by preventing NP aggregation [15,16]. The therapeutic potential of polymeric NP generally depends on their physicochemical properties such as size, shape, zeta potential, loading capacity, and surface functionalization with suitable surfactants [17,18].

### 2.1. Background

Soap (general formula RCO-ONa) is formulated from anionic surfactants, and the first records of its manufacture date back to 2800 B.C. in ancient Babylon [19,20]. However, the word “surfactant” was first used in the 1940s [3,17]. More recently, in the 1960s, the term “amphiphilic” was introduced by Paul Winsor, a word that comes from two Greek roots (Amphi meaning “double”, and Philos meaning “affinity”) [3]. Finally, between the 1950s and 1970s, the first models based on n-alkylammonium were developed to study the arrangement and orientation of cationic surfactants in solid interfaces; these explain the position and approximate inclination angle of adsorbed surfactant molecules and their physicochemical implications in the surface coating [17,21]. As a result, the surfactants industry has increased due to its wide application and discoveries. A field in which the utility of surfactants is currently exploited is the pharmaceutical industry since scientists have developed polymeric NP to administer therapeutic and diagnostic agents [22].

### 2.2. Stabilization Mechanisms

The physical stability of NP mainly depends on electrostatic, steric, entropic, and Van der Waals forces [23]. The DLVO theory (Derjaguin-Landau-Verwey-Overbeek) describes the interaction energy between particles as the sum of electrostatic and Van der Waals forces; the resulting equilibrium explains the stability (suspension or flocculation) of colloidal systems [24]. When the surface charge of NP is homogeneous (either positive or negative), the Van der Waals and electrostatic forces oppose each other, causing the net force between particles to be strongly repulsive, and a stable suspension is formed [24,25]. As NP get closer to each other, their ionic atmospheres begin to overlap, and a repulsive force develops. On the other hand, Van der Waals interactions between NP are also generated due to forces between individual molecules in each colloid [26].

More stable dispersions can be obtained when the system contains oppositely charged NP and surfactants, such as anionic NP and cationic surfactants or vice versa. The dominant mechanisms are electrostatic interactions and hydrogen bonding [27]. In electrostatic stabilization, a minimum zeta potential of |20 mV| has been suggested [28]. However, there have been reported cases in which nanosuspensions with zeta potential below |20 mV| are physically stable [29,30]. This could be explained by the addition of non-ionic surfactants and the resulting steric effect. Therefore, the interpretation of the zeta potential to predict the stability of colloidal nanosuspensions should be considered with caution and in conjunction with the surfactants utilized [29].

### 2.3. Ionic and Non-Ionic Surfactants

Surfactants are classified according to the charge of their main group (polar head): non-ionic (uncharged) and ionic (charged) (Figure 1c). Among those that are charged, we find anionic (negatively charged), cationic (positively charged), and amphoteric (both positively and negatively charged) [31]. The charges of the zwitterionic or amphoteric surfactants can be permanent or can depend on the pH value to which they are exposed; for example, betaines can function as cationic surfactants at highly acidic pH [32,33]. A study showed the sensitivity of sulfobetaine to alteration of pH and inorganic salt. Hydrogen bonds are formed between the amide groups of 3-(*N*-erucamidopropyl-*N*,*N*-dimethyl ammonium) propane sulfonate and coordinated water in trans-[FeCl_2_(H_2_O)_4_] Cl structure; this masks the ionic forces of repulsion between the head groups of surfactants [34]. The zwitterionic head groups of phosphatidylcholine show electroneutral charges and high hydration, making them highly stable in aqueous media [35]. In addition, their electrostatic attraction causes the polarity of ionic surfactants to the dipoles of water. Moreover, non-ionic surfactants are solubilized without ionizing through the effect of weak hydrophilic groups such as ether-type bonds and hydroxyl groups; these are employed more frequently in pharmaceutical products. In Table 1, examples of surfactants commonly used in pharmaceutical formulations are mentioned.

### 2.4. New Surfactants

A wide range of classic surfactant agents is based on alkyl, peptides, lipids, DNA, molecular ligands, and polymers [17]. In recent years, particular interest has been placed in developing new biocompatible surfactant agents, representing low toxicity for the environment and human use; some of these agents are mentioned below. Carbohydrates: Have been studied due to their biodegradability and low toxicity profile. Smulek et al. [39] investigated a series of alkyl glycosides containing d-lixose and 1-rhamnose with alkyl chains of 8–12 carbon atoms. The results revealed that long-chain alkyl glycosides could be inexpensive biocompatible surfactants. Alkylpolyglucosides: Include a group of non-ionic surfactants with excellent wetting, dispersing, and surface tension reducing properties; their use for the stabilization of lipid NP is more frequent than classical stabilizers [40]. ImS3-n (3-(1-alkyl-3-imidazolium) propane-sulfonate): Represent a versatile class of zwitterionic compounds, which form normal and inverse micelles, capable of stabilizing NP in water and organic media [41]. Polyhydroxy Surfactants: Involve ethylene oxide-free non-ionic stabilizers known for their dermatological properties and favorable environmental profile [42]. Rhamnolipids: Biosurfactants produced by marine bacteria have shown a lack of cytotoxicity and mutagenicity, which justifies their commercial exploitation as natural and ecological biosurfactants [43,44]. Animal-derived surfactants: in the same context of using biocompatible surfactants, bioglycolipids such as cerebrosides (which represent a group of non-ionic surfactants) and gangliosides (these are good cationic surfactants) have been proposed [45]. PEG-ylated amides: PEG-conjugated amides improve the stability of nanosystems and allow a prolonged circulation time, reducing the phenomenon of accelerated blood clearance [46,47]. Recently, BioNTech and Pfizer used two novel surfactants in the formulation of their BNT162b2 mRNA Covid-19 vaccine, the PEG-ylated lipid ALC-0159 (2-[(polyethylene glycol)-2000]-*N*,*N*-ditetradecylacetamide) and the cationic lipid ALC-0315 ((4-hydroxybutyl)azanediyl)bis(hexane-6,1-diyl)bis(2-hexyldecanoate)) [48,49]. ALC-0159 allows forming a hydrophilic layer that sterically stabilizes the nanosystem, contributing to storage stability and reducing non-specific binding to proteins. Furthermore, ALC-0315 forms an electrostatic interaction with the negatively charged RNA skeleton allowing its stabilization, encapsulation, and the formation of particles [48,50]. Notably, several already known surfactants have been associated with new biological activities such as ceramides. For example, exogenously administered N-hexanoyl-D-erythrosphingosine has been reported to arrest the cell cycle, and in combination with Paclitaxel in biodegradable polymeric NPs can significantly enhance apoptosis in multidrug-resistant and sensitive cells [51]. Other strategies include stabilizing solid micro- or NP (Pickering stabilization), surfactant-free, and confers high resistance to coalescence, making it attractive for pharmaceutical applications, where some surfactants can cause adverse effects [52,53]. In addition, organic and inorganic particles are used, utilizing steric and/or electrostatic repulsion to inhibit coalescence and improve emulsion stability. A recent study reported Pickering emulsions stabilized by biodegradable poly(lactic-co-glycolic acid) (PLGA) NP and exposed that the degree of stabilization is highly dependent on the polymer composition [54].

## 3. Polysorbates

PS are one of the most utilized stabilizers in the industry. Their physicochemical properties have a striking impact on nanoparticulate systems. Despite its broad applications, new PS properties and modes of employment are still being found and reported nowadays.

### 3.1. Physicochemical Properties

Tween is the commercial name for a group of compounds based on PS. PS are amphiphilic molecules synthesized by the reaction between sorbitan fatty acid ester with ethylene oxide. The PS chemical structure has a sorbitan head group where the hydroxyl groups are bound to a polyethylene glycol (PEG) chain. Basically, in the PS structure, the fatty acid side chain (hydrophobic fragment) is esterified with one of the PEG (hydrophilic fragment) side chains. However, the length of the PEG chains, the esterification in one or more hydroxyl groups in the side chain, changes in the head group, and the fatty acid composition are the fingerprint of each type of PS [55,56].

The HLB of PS (between 9.6–16.7) combined with their low critical micelle concentration (CMC) gives the PS high surface activity, even at low concentrations. Therefore, PS confer high stability to the NP during storage in an aqueous medium and freeze-drying or freeze/thaw processes. In addition, the use of PS prevents interface-induced aggregation and surface adsorption in particulate systems. Therefore, a PS concentration of 0.001 to 0.1% (*w*/*v*) is commonly utilized in the biopharmaceutical area with suitable performance. PS are degraded through enzymes and chemically by autoxidation or hydrolysis pathways. The susceptibility to the oxidation process in PS resides in the PEG ester bonds and unsaturated alkyl chains. On the other hand, the chemical PS hydrolysis is catalyzed in acid or basic conditions and directly affects the free fatty acids and the unesterified sorbitan [33,34,35].

### 3.2. Types

The commercial PS present more complex interactions between their molecules. In addition, the extensive subproducts and isomers formed during the synthesis process can affect the stability and chemical activity. The group of PS accepted for the formulation in human products comprises the PS-20, PS-60, PS-65, and PS-80. However, in cosmetics, the PS-21, PS-40, PS-61, PS-81, and PS-85 are also used. All the PS are hydrophilic emulsifying and stabilizing agents. The distinct levels of lipophilicity and hydrophilicity among the PS arise from the chemical structural differences. The PS-20 presents the lauric acid as the main fatty acid, making it a more hydrophilic molecule than the PS-60 and PS-80, which present larger fatty acids: stearic acid and oleic acid, respectively. Likewise, PS-65 has stearic acid as the primary fatty acid; however, PS-65 possesses three esterified hydroxyl groups with stearic acid, which confer more lipophilicity (Figure 2) [55,56,57].

### 3.3. Uses

PS are widely employed in the food, cosmetic, and pharmaceutical industries. Their roles include oil/water emulsifier, detergent, dispersing agent, solubilizer, and stabilizer in cosmetics. PS are common ingredients in applications for skin, hair, nails, and mucous membranes, with a typical application several times a day depending on the product. With a medical approach, the PS-20 and PS-80 are listed as clarifying agents in ophthalmic products and as cleaning, wetting, or solvent agents for contact lenses in concentrations below 1.0% [58,59]. Nowadays, PS offer a wide application in the development of nanoparticulate systems to improve drug physicochemical properties, bioavailability, and therapeutic enhancement (Table 2).

### 3.4. Derivatives

PS are mainly utilized to refine the physicochemical properties of nanoparticle systems. With this scope, the synthesis of new derivatives of PS has also been investigated. For example, Masotti and coworkers [83] synthesized three PS-20 derivatives, differenced each other by substituted head groups. The PS-20 was functionalized with glycine, *N*-methyl-glycine, or *N*,*N*-dimethyl-glycine to develop a pH-sensitive system. PS-20 derivatives were obtained by reaction of the PS with different amino substituents in the presence of H_2_SO_4_ at 90 °C for 12 h. The three derivatives exhibited the capacity to form vesicles complex with cholesterol molecules—the vesicles presented a size between 176 and 320 nm, in a pH range of 5.5 to 7.4. The three types of vesicle size increased while the pH decreased. In contrast, the Z potential decreased to more negative values while the pH increased. These changes confirm the pH-sensitive effect of the vesicles prepared with the modified PS and open the opportunity to develop new systems with pH response activity.

Similarly, other groups have modified polymers using PS to improve some functions in nanoparticle systems. These molecules are not accurately derivatives of PS, but the addition of PS improved cellular uptake, even in multidrug-resistant cancer cells [66,67]. Evidence of this is the PLGA-PS-80 copolymer NP developed by Yuan et al. [67]. The authors synthesized by esterification reaction 0.3 mmol of PLGA-COOH with 0.6 mmol of PS-80. The novel copolymer was employed to obtain NP by the nanoprecipitation technique. The particles presented a size of 156.5 ± 8.6 nm, a PDI of 0.14, and a zeta potential of −15.4 ± 1.1 mV. PLGA-PS-80 NP were able to load 5% of 1 mg paclitaxel. PLGA-PS-80 NP increased the cellular uptake in lung cancer cell line A549/T, with a higher level of cytotoxicity than unmodified PLGA NP. Furthermore, the nanoparticle system was evaluated in vivo and exhibited a higher antitumor efficacy than free taxol. The development of new derivatives and copolymers of PS opens the opportunity to improve existing drugs and their applications.

### 3.5. Examples of NP Applications

The PS-20 and PS-80 are mostly applied in biopharmaceutics to stabilize polymeric NP, and both have a suitable preventing protein adsorption and low toxicity profile [55,56] (Table 2). However, the PS-80 presents a longer monounsaturated chain, making them more surface-active with a lower CMC. This property renders the PS-80 the most utilized PS for nanoparticle systems. Moreover, PS-80 is reported as a molecule with the functionality to enhance the crossing of NP through biological barriers. For example, the Poly(n-butyl cyanoacrylate) (PBCA) NP coated with PS-80 improved the dalargin-induced analgesia. Furthermore, PBCA NP summited to double PS-80 coating could cross the gastrointestinal barrier after oral administration [69,84]. Similarly, the presence of the PS80 on the nanoparticle surface improves the crossing of the blood-brain barrier (BBB). The PS chemical properties interact with plasma protein such as apolipoprotein E. Receptors for the apolipoprotein expressed in the cells of the BBB promote receptor-mediated endocytosis; in consequence, the NP treated with PS-80 enhance the chance of the drug to reach the brain [85,86].

Interestingly, the PS can be employed alone to form micellar systems to carry drugs. For example, Ravichandran et al. [87] developed a PS-80 nanomicellar system to transport piperlongumine and indocyanine green for cancer treatment. The system improved the drugs’ storage stability, increased the photothermal conversion and cellular uptake of indocyanine green. In addition, the pro-oxidant activity of the piperlongumine was maintained, which was evidenced by the increased levels of reactive oxygen species in MCF-7 cell cultures. This research proves the effective use of PS and opens a great field to research the PS functionality. More examples are represented in Table 2.

### 3.6. Toxicity in NP

PS have occasional reports of hypersensitivity following their topical and intramuscular administration [59,88]. However, the PS concentrations are crucial to establish biosecurity in their application. For example, Elmowafy et al. [75] reported a cell death of around 70% in cells exposed to polycaprolactone (PCL) NP stabilized with PS-80 at high concentration (100 µg/mL). Contrariwise, the same NP stabilized with PS-80 at low concentration (1.56 µg/mL) did not present cytotoxicity.

However, high cytotoxicity can be helpful when the nanoparticle application involves one type of cancer. For example, Nguyen et al. [64] developed PLGA NP to entrap artesunate (an anti-malarial agent) and tested the anticancer effect. The nanoparticle system presented a particle size of around 150 nm with PS-80 as stabilizer, with a 23.6 ± 0.61% of drug loading capacity. The cytotoxicity of the artesunate entrapped in nanoparticulate systems was higher than the artesunate alone in SCC7, A549, and MCF7 culture cells. Contrariwise, when the aim is to improve drug therapy, high cytotoxicity can be a problem, and the particles’ redesign is necessary. For example, Chang et al. [89] developed a PLGA NP system to enhance the transport throughout the BBB and evaluated the effect of PS-20. In this study, the authors assessed the cytotoxicity following the tight junction aperture. The authors described the high toxicity of the NP stabilized with PS-20, evidenced by a high sucrose endothelial permeability coefficient Pe > 2 × 10^−3^ cm/min after 1 h of incubation. This test was crucial to eliminate the use of the PS-20 NP in the subsequent experiments.

### 3.7. Advantages and Disadvantages

The physicochemical properties of PS make them a great stabilizer in nanoparticle systems. For example, PS concentration can directly adjust the size of the adequate nanoparticle. The use of PS also confers high stability in the formulation process as well as in biological systems. Furthermore, PS increase the NP’s bioavailability by decreasing the interaction with plasma proteins and increasing the cellular uptakes to cross biological barriers [79].

However, the concentration of PS in NP should be formulated carefully because high concentrations can alter the fluidity of barriers and can occur hypersensitivity reactions. Inversely, the use of low concentrations could compromise the efficacy or stability of the NP during storage. Therefore, the development of analytical methods that quantify the amount of PS present on the surface of the NP takes relevance. These methods aim to monitor and solve the optimum amount of PS [61,90,91,92], avoiding undesirable reactions.

In certain circumstances, the pharmaceutical storage and manufacturing conditions could catalyze the PS hydrolysis and autooxidation, compromising the NP’s stability [57]. Moreover, the lack of uniformity in commercial PS is caused by monoesters and polyesters of polyoxyethylene (POE), and POE sorbitan and POE isosorbide fatty acid esters, which are typical products during the synthesis reactions. The inevitable presence of these products affects the PS final chemical composition and performance in chemical, biological, or mechanical stimuli. Although the variation is not exclusive to specific suppliers, the variation among batches is common [93].

## 4. Polyvinyl Alcohol

PVA is a linear synthetic, amphiphilic, semicrystalline, biocompatible, biodegradable, highly flexible, and nontoxic polymer that functions as an emulsifying agent by lowering the solutions interfacial tensions [94,95]. Its capacity to form relatively small particles and uniform size distribution makes it an appropriate candidate for biomedical applications [96,97,98,99]. In addition, PVA stabilizes emulsions because a fraction of PVA remains associated with the NP by forming an interconnected network with the polymer at the interface [100]. PVA is produced only by indirect methods [101,102,103], generally by hydrolysis or methanol transesterification (methanolysis) of polyvinyl-acetate (PVAc) [104,105], and the result is a hydrophilic polymer with a simple structure having a pendant hydroxyl group [95] that is resistant to many solvents [106]. In this respect, PVA is soluble in highly polar and hydrophilic solvents, such as Dimethyl Sulfoxide (DMSO), *N*-Methyl Pyrrolidone (NMP), Ethylene Glycol (EG), and water (the most important solvent of PVA) [107]. On the other hand, PVA is resistant to oil, grease [108], and some solvents such as ethyl acetate, acrylonitrile, acetonitrile, and ammonia [109,110]. The resistance to most organic compounds and solvents can be advantageous; for example, PVA-based materials may protect packaged products from secondary contamination by those solvents [108].

### 4.1. Physicochemical Properties

The physicochemical and mechanical properties of PVA vary according to the degree of polymerization (DP) and degree of hydrolysis (DH) (see Section 4.2). Differences include water-solubility, adhesion and mechanical strength, gas barrier and aging resistance, thermostability, chemical resistance, film-forming ability, low fouling potential, pH stability, viscosity, high polar character, and easy processability [111,112,113,114]. PVA has a white to yellowish color [101]. The melting point is 228 °C for fully hydrolyzed grades and 180–190 °C for partially hydrolyzed grades [115], due to the content and distribution of acetyl groups, tacticity, and water content [101]. The principal solvent of PVA is water [116], and the HLB value is 18 [117].

### 4.2. Types

PVA is generally derived from PVAc through partial or total hydrolysis to remove the acetate groups; then, there are types of PVA according to DP and DH [105,106,115,118,119]. PVA is commercially available in grades according to DH and viscosity. Partially hydrolyzed grades range from 84.2 to 89.0%, moderately hydrolyzed from 92.5 to 96.5%, and the completely hydrolyzed from 98.0 to 99.0% [94,101,119,120]. PVA hydrolyzed at 40 and 80% have a molecular weight of 72,000, and 9000–10,000, respectively [94,119]. There is an inverse relationship between molecular weight and DH. Lower molecular weight results in low viscosity, less aqueous solubility, and high flexibility, while lower DH results in increased solubility, flexibility, water selectivity, and adhesion to hydrophobic surfaces [108,120].

### 4.3. Uses

PVA is the world’s most-used synthetic polymer and is included in the Handbook of Pharmaceutical Excipients [115]. Its versatility and properties have attracted interest from several industrial applications [94]. In medicine, PVA is convenient because it has low protein absorption and chemical resistance [121]. Thus, it has been used for applications such as controlled drug delivery systems [106], soft contact lenses [96,106], eye drops [122], embolization particles [123,124], tissue adhesion barriers [125,126], transdermal patches and jellies for the skin [115], and as artificial cartilage and meniscus [106,127]. Moreover, PVA helps dissipate heat and prolongs electronic devices’ lifetime [128], is helpful for wastewater treatment [129,130,131,132], as well as in the industrial, commercial, and food sectors [106,133,134,135,136,137,138,139,140].

### 4.4. Derivatives

As mentioned, PVA is very versatile, and with some physical or chemical modifications, its properties can be modulated to improve its performance as a drug targeting and stabilizer, among other benefits [120]. For biomedical purposes, physical modifications of PVA like freeze-thawing [141], annealing [120,142], irradiation [143], and composites [144,145] are preferred to avoid any possibility of toxic residues [139]. In addition, these modifications cause molecular rearrangement by forming more crystalline regions [120].

Chemical modifications by crosslinking to PVA can be done through different techniques due to its hydrophilic characteristic. In theory, it can be done using any compound capable of reacting with -OH groups [132,146] such as aldehydes, dialdehydes like glutaraldehyde [147,148,149], carboxylic acids (lactic acid, maleic acid, sulfosuccinic acid) through esterification [146,150,151,152], sodium tetraborate [153], epichlorohydrin [154,155], and carboxymethyl [156] among others. These modifications alter physical and mechanical properties, reducing water solubility, and increasing polymer rigidity and chemical stability [120,132,157,158,159,160].

### 4.5. Examples of NP Applications

PVA is highly effective as a colloidal protector and stabilizer of suspensions. For example, polymer nanolatexes are obtained through emulsion polymerization. The use of PVA as the sole stabilizer helped obtain an adjustable NP size from 60 to 100 nm with a PDI between 0.05 to 0.07 [161]. PLGA-PVA NP loaded with chitosan-dextran sulfate-doxorubicin were designed and successfully delivered doxorubicin to MCF-7-DOX-R cells [162], obtaining the desired anti-proliferative effects. A poly(acrylic acid) (PAA)-b-PVA double-hydrophilic block copolymer, with a pH- and IS-responsive block (PAA) was used to stabilize γ-Fe_2_O_3_ NP, improving its colloidal stability for its potential application for remotely magnetically triggered drug release to some tumor site [163]. Eudragit^®^ L100 NP loaded with ketorolac tromethamine with a size of 153.7 nm, a PDI of 0.318, a zeta potential of −16.9 mV, and DL of 36.3% were successfully incorporated in PVA-hydroxyethyl cellulose (HEC) films for ophthalmic drug delivery system [164]. More examples are presented in Table 3.

### 4.6. Toxicity in NP

Sprague-Dawley rats were fed with PVA (2000, 3500, and 5000 mg/kg/day) for up to 90 days, and no toxicological effects were observed. When tested in NP, similar results were obtained. PLGA/PVA NP tested in Human-like-THP-1 macrophages found less cytotoxicity at high concentration (1 mg/mL) than other stabilizers [217]. Polymeric NP of chitosan-g-poly(methyl methacrylate) (PMMA) and PVA-g-PMMA were ionotropically crosslinked with sodium tripolyphosphate to mask the positive charge and successfully avoid toxicity after 0.8 mg/kg intranasal administration of NP in Hsd:ICR mice [218]. In addition, tamoxifen-loaded-PLGA/PVA NP exhibited low toxicity in C127I cells (up to 10 μg/mL) and rats with 12-dimethylbenz(a)anthracene-induced breast cancer (3 mg/kg) [219]. Paclitaxel-loaded PVA-g-PLGA NP tested in RbVSMC cells presented a 30% reduction in cell viability at 300 μg/mL, while biocompatibility was confirmed to 370 μg/mL in drug-free NP [220].

### 4.7. Advantages and Disadvantages

The high biodegradability in the environment may be the most desirable characteristic of PVA because it can easily be degraded by bacteria (Gram-negative and Gram-positive) and *Penicillium sp* [102,105,115,140]. Furthermore, PVA-coated NP exhibit a low level of non-specific interaction with solutes like cell adhesion proteins due to its hydrophilic nature [221], which could be beneficial or harmful. On the other hand, it has been reported that a fraction of PVA forms a stable matrix on the polymeric surface that cannot be removed, affecting the physical properties of NP and their interactions with the surrounding environment [222]. Finally, a problem with the reports of PVA-coated NP is that the molecular weight and the hydrolysis percentage are generally not reported, affecting the reproducibility of the studies.

## 5. Poloxamers

Poloxamers, also known as Pluronic^®^, are tri-block copolymers with amphiphilic properties, which can be found in three different forms: liquid, paste, and flake. They were commercially first produced by BASF Corporation in 1950 [223,224]. These block copolymers contain two blocks of the hydrophilic POE separated by one block of the hydrophobic poly(propylene oxide)(PPO) in an arrangement A-B-A (POE-PPO-POE) [225]. The possibility of modifying the amount and relation between POE and PPO in the poloxamer structure allows the obtention of different physicochemical characteristics. Poloxamers present surfactant properties and are widely utilized in the stabilization of nanostructured systems. Furthermore, these copolymers can interact with cells and cell membranes, providing a high potential to be applied in the design of new biomaterials and novel nanomedicines (Figure 3) [226,227].

### 5.1. Physicochemical Properties

Poloxamers are synthesized by sequential polymerization of ethylene oxide and PPO units in sodium hydroxide and potassium hydroxide. Their chemical formula is HO[CH_2_CH_2_O]*_x_* [CH(CH_3_)CH_2_O]*_y_* [CH_2_CH_2_O]*_x_*OH, where *y* is higher than 14. Depending on the *x* and *y* values, these copolymers present different amphiphilic properties based on their HLB values. The physicochemical characteristics of poloxamers could be modified by changing the POE and PPO relation mass, obtaining different sizes, lipophilicity, and hydrophilicity. Regularly, the molar mass ratio between POE and PPO blocks ranges from 8:2 to 9:1, derived from the coexistence of hydrophilic and hydrophobic sections in the poloxamers structure. These copolymers present a high temperature-dependent self-assembling characteristic as well as thermo-reversible properties. Solutions of poloxamers with concentrations higher than their CMC produce gels at temperatures above their sol-gel transition temperature. Additionally, PPO blocks dehydrate at high temperatures, and it is less soluble in water, which triggers the formation of the micelles with the dehydrated PPO core and hydrated POE shell. Consequently, depending on the main properties needed for the application, appropriate physical and chemical modifications could be applied in the poloxamer synthesis. Interestingly, several reports demonstrated that poloxamers could decrease the cell membrane microviscosity (membrane fluidization) due to the presence of hydrophobic PPO chains [223,228]. Furthermore, it has been reported that poloxamers could reduce multidrug resistance, inducing a dramatic reduction in ATP levels in cancer and barrier cells, and inhibiting drug efflux transporters [227,229,230].

### 5.2. Types

As we mentioned, poloxamers are polymorph materials covering a range of gelation states from liquid to paste and solid, depending on the molar mass ratios between the POE and PPO blocks [224]. This class of A-B-A copolymers offers a pool of more than 50 materials with molecular weights ranging from 1600 to 14,600 Da that present different properties [231]. In the commercial nomenclature of these copolymers, the name is composed by a letter that indicates the morphism of each copolymer: liquid (L), paste (P), and flake (F), followed by two or three digits, which is related to their structural parameters [224,232]. For example, the Pluronic^®^ known as L121 (poloxamer 401) has a liquid presentation with above 2,000 Da of molecular weight and HLB of 3; meanwhile, Pluronic^®^ F68 (poloxamer 188) has an average molecular weight of 8,400 Da, a percentage of POE around 80%, and an HLB value of 29 with flake morphology [231]. On the other hand, Pluronic^®^ P123 (poloxamer 403) is a paste with an HLB value of 8 and an average molecular weight of 5,750 Da [224,233].

### 5.3. Uses

The structural versatility of poloxamers makes them particularly attractive emulsifying, solubilizing, and dispersing ingredients for pharmaceutical formulation [234,235]. It has been reported that the incorporation of low molecular weight drugs into poloxamer micelles could increase drug stability and solubility, improving the pharmacokinetics and biodistribution. These polymeric micelles have been employed for the oral administration of tumor-specific and brain drugs [234,236,237]. In addition, these copolymers have been used as food additives. Moreover, due to their thermosensitivity, poloxamers can form hydrogels, injectable hydrogels, 3D scaffolds, micro/nanofibers, and cell carrier constructs; consequently, these copolymers have been utilized in tissue engineering and biomaterials fields [232,238,239,240].

### 5.4. Derivatives

The synthesis of chemically cross-linkable poloxamers has also been analyzed to enhance their mechanical properties. The chemical structure of poloxamers presents only reactive groups available for the modification at the end of the chains; thus, chemically cross-linkable groups can only be used to end-cap the triblock chain. Ethoxylsilane and methacrylate/acrylate are two groups employed for the crosslink of end-capping groups [241]. To introduce ethoxy silane end-capping group, (3-isocyanatopropyl) triethoxysilane can be utilized to react with the hydroxyl groups of poloxamers under the catalysis of 2-ethyl-hexanoate. For the coupling of methacrylate/acrylate, methacryloyl chloride/acryloyl chloride reacts with hydroxyl groups on both ends, resulting in higher mechanical properties [241,242]. In this context, in 2021, Popescu et al. developed a hydrogel from a natural polymer and poloxamer 407 obtained by thiol-acrylate photopolymerization to be employed as a wound dressing [243].

### 5.5. Examples of NP Applications

Poloxamers have been widely applied in nanotechnology as a stabilizer and nanoparticle shell component for different applications. In 2019, Del Prado et al. developed a system of PCL NP stabilized by poloxamer 188 [244]. The results suggested that the presence of poloxamer in the nanoparticle’s surface produced a stable nanodispersion during six months of storage. Interestingly, the nanocarrier leads to a decrement in reactive oxygen species, which the authors attributed to the presence of poloxamer 188 [245]. Similarly, in 2021, the poloxamer 188 was employed as a stabilizer of rivaroxaban-loaded PLGA NP as a novel strategy for treating thrombotic disorders [246]. The nanosystem presented a spherical morphology with an average size of 200 nm and PDI of 0.09, the latter suggesting a homogenous size distribution. Another poloxamer highly utilized as a stabilizer is the poloxamer 407. Recent research reported the evaluation of amphotericin-loaded PCL NP using poloxamer 407 as a surfactant [247]. The spherical nanocarriers presented a mean hydrodynamic particle size of 183 nm and encapsulation efficiency of 85%. The elaboration of poloxamer 407-based NP also has been explored [248,249]. Another research reported NP of poloxamer 407 with a size around 100 nm and PDI of 0.122; the formulation exhibited appropriate properties to deliver chemotherapeutic agents [248]. Table 4 compiles examples of poloxamers formulated in nanoparticulate systems.

### 5.6. Toxicity in NP

Several studies have reported the use of poloxamers-based or -coated NP in cell culture, finding that the toxicity of these surfactants is relatively low [276]. For instance, Li et al. [277] reported that after 14 days of feeding with solutions of poloxamer 235/poly (lactic acid) NP at different concentrations, no deaths or treatment-related complications were observed in mice, even in the higher concentration treatment. In 2017, the enhanced viability of pancreatic islets due to nanosystems based on poloxamer 407/chitosan and bilirubin entrapped into NP was reported [278]. In other work, a nanostructured system was developed based on poloxamer 407, and poloxamer 403 modified in the end chains with vitamin E succinate to encapsulate paclitaxel [279]. The cell viability was evaluated in bone marrow-derived macrophages and a human glioma U87 cell line. The NP presented no significant changes in viability of macrophages and high cytotoxicity in human glioma U87 cells, which was related to the therapeutic effect of the NP.

### 5.7. Advantages and Disadvantages

As mentioned, poloxamers exhibit characteristics that are very useful for the pharmaceutical and biomedical fields. For instance, thermosensitivity, high capacity to solubilize drugs, drug release properties, and the absence of toxicity in mucous membranes [240,280]. Furthermore, these safe materials present the ability to inhibit drug efflux transporters, representing an essential advantage for drug release systems. On the other hand, a disadvantage of these block copolymers is their fast degradation rate in vivo [281]. Additionally, it has been reported that Pluronic^®^ copolymers presented low cytotoxicity and, remarkably, weak immunogenicity in topical and systemic administration. It is known that POE-PPO-POE copolymers are non-degradable; however, molecules with a molecular weight of 15 kDa or less are usually filtered by the kidney and cleared in urine [282]. On the other hand, in recent publications, the complement system activation by POE-containing polymers has been analyzed [283,284]. Authors suggested that these polymers could trigger acute hypersensitivity reactions or pseudoallergic reactions [283]. However, some evidence demonstrated that this effect is observed only in highly responsive patients to complement activation [285]. Consequently, the Pluronic coating must be carefully developed.

## 6. Influence of Non-Ionic Surfactants on the Interaction with Biological Barriers

The impact of surfactants on the physicochemical parameters of NP is not limited to stability phenomena. There is a significant influence of surface phenomena driven by surfactants and interactions at the cellular level [8,286] (Figure 4). Therefore, it is possible to increase or decrease the interaction with cells according to the type of surfactant; even the vectorization process can be favored in a certain way. The influence of non-ionic surfactants on the passage through the main biological barriers involved in the pharmacokinetic processes is described below.

### 6.1. Blood-Brain Barrier

The BBB is a highly sophisticated brain barrier with tight junctions between endothelial cells and a foreign substance detection system; therefore, it represents a challenge for drug passage. Interestingly, Voigt et al. [18] conducted a blood-retina barrier passage study as a BBB model of fluorescent PBCA NP with different types of surfactants: non-ionic (PS-20, PS-80, polyethylene glycol monododecyl ether, poloxamer 188), anionic (sodium dodecyl sulfate (SDS), and cationic (dextran). The authors used real-time imaging of retinal blood and in vivo confocal neuroimaging during and after nanoparticle injection. The study revealed that non-ionic or even cationic surfactants allowed a successful BBB passage, while particle size and zeta potential had no influence. Furthermore, even when the authors decreased the size of the NP to 87 nm but added SDS to the non-ionic surfactant, they did not observe crossing in the BBB [18].

### 6.2. Intestinal

Historically, the oral route has been the preferred drug administration route due to patient comfort, ease of application, and low treatment costs. However, there are different limitations inherent to the gastrointestinal region that produce variations in the bioavailability of drugs. The use of NP is desirable to ensure adequate bioavailability, drug stability, and even control in the sustained release system. The presence of non-ionic surfactants can increase mucus penetration, reduce recognition and clearance, enhance plasma circulation times, and promote drug accumulation. PEG is one of the predominant strategies, and it is also a passive mucopenetrating excipient that reduces interactions with luminal components and mucus in the gut [287].

### 6.3. Intranasal

Nasal administration is intended for local or systemic action. Some advantages of the nasal region include a large surface area, low enzymatic activity, vascularized subepithelial layer with direct passage to the systemic circulation, and evasion of the first-pass metabolism in the liver. On the other hand, some challenges include low membrane permeability of polar drugs and rapid clearance [288]. In this respect, non-ionic surfactants as absorption enhancers may play an important role. For example, poloxamer 188 gels promote the permeation of nanocubic vehicles and PLGA mixture-based DNA NP. Furthermore, an intentional comparative study to evaluate the effect of non-ionic surfactants on the intranasal permeation of sumatriptan succinate demonstrated that Laurate sucrose ester promoted higher absorption and absolute bioavailability. However, the effects of polyethoxylated castor oil (cremophor EL^®^) and poloxamer 188 were also desirable [288].

### 6.4. Pulmonary

Pulmonary drug delivery allows local and systemic effects. The lung has advantages such as avoiding the gastrointestinal environment and reducing the first-pass metabolism of drugs. However, its main barriers in the absorption process are the epithelial and capillary cell barrier and a surfactant layer. Strategically, one of the tools in drug delivery is the decrease in particle size and surfactants at the interface [289]. In this regard, non-ionic surfactants act as modifiers of the absorption of drugs in the lung. For example, the combination of PS-80 and poloxamer 407 increased the lung area under the curve of itraconazole particles up to nine times through a wetting mechanism with the absence of pro-inflammatory components. A similar strategy utilized PEG and PVA to stabilize sebacic acid particles obtained by an emulsion method. Furthermore, poloxamer 188 can also be used to stabilize inhalable particles with the additional advantage to prevent the absorption of proteins and peptides that can be absorbed in the air-liquid interface of droplets and produce surface erosion [289].

## 7. Quality by Design in the Choice of a Surfactant: The Royal Road

The pharmaceutical industry is working hard to achieve robust and high-quality drug products. The QbD, which the International Conference of Harmonisation of Technical Requirements for Registration of Pharmaceuticals for Human Use defines in the harmonized tripartite guideline for the pharmaceutical development Q8(R2) [290], is “a systematic approach to development that begins with predefined objectives and emphasizes product and process understanding and process control, based on sound science and quality risk management.” The QbD has been extended to the systematic development of drug products by minimizing challenges, including a lack of consistency in quality and product robustness. The application of QbD to nanopharmaceutical products has several benefits for optimizing product performance in terms of complex design, dynamic material properties, and stringent regulatory requirements for quality attributes (QAs), including particle size, zeta potential, drug loading, in vitro drug release profile, surface morphology characteristics, pharmacokinetic performance, drug stability, and impurity profiling [291].

In the current Quality by Test (QbT) system, product quality is ensured by following a sequence of steps, including raw material testing, fixed drug product manufacturing process, and end-product testing. If the specifications or other standards are met, the product may be kept in the manufacturing or incorporated into the market. Otherwise, it will have to be reprocessed [292]. Due to this situation, several compounds have been studied utilizing QbD. The procedure for implementing QbD in the suspension of NP includes the following steps (Figure 5): (1) determine the stabilizers and preparation method according to the Quality Target Product Profile (QTPP), (2) define the Critical Quality Attributes (CQAs) (particle size, charge, stability, etc.) and from that to establish the critical material attributes (CMAs) and the Critical Process Parameters (CPPs) based on prior knowledge when conducting risk assessment, and (3) conduct Design of Experiments (DoE) to build a design space and verify its feasibility and robustness [293]. DoE is a better strategy than changing a single experimental factor and keeping other factors constant that can lead to more experiments than are feasible, especially if many variables are of concern. Furthermore, this eliminates the possibility of evaluating factor interactions [294]. From careful and systematic considerations, the industry and researchers can assess the influence of variables (for example, type and amount of stabilizer) on the nanoparticle CQAs, which helps achieve minimal particle size, good crystallinity, a high yield percentage, and more [295].

Based on the above, it can be considered that the search for a successful nanoparticle-based product results from the effect of several variables. One of these variables is associated with the selection of the stabilizer since it has been shown in several studies that the stabilizer, which is generally a surfactant, has a significant impact on several response variables of the NP; for example, on the efficiency of encapsulation, particle size, charge, among others. Through the employment of QbD, Saha et al. [296] indicated that the encapsulation efficiency of resveratrol in mucoadhesive lecithin/chitosan NP for prolonged ocular drug delivery was significantly influenced by the concentration of poloxamer 407 and also revealed a significant interaction with the concentration of resveratrol utilized in the manufacture of their NP. The evaluation of the effect of poloxamer 407 was established after a risk analysis in which the polymer, together with the lecithin concentration, chitosan to lecithin ratio, and drug concentration, were categorized by the severity score as high-risk material attributes. The poloxamer 407 was ranked with a higher risk score than other concerning parameters of the manufacturing process, such as the molecular weight of chitosan, lecithin grade, type of needle, stirring speed, and injection rate others. Moreover, parameters such as particle size, Z potential, encapsulation efficiency, and drug release were considered QAs of greater relevance.

On the other hand, Patel et al. [297] evaluated the effect of the type of surfactant (poloxamer 188 and PS-80) and concentration (0.5%, 2.0%, and 5.0%) on the development of topical arginine solid lipid NP (SLN) from a QbD approach. After a risk analysis, the authors found that the surfactant concentration presented a higher risk priority number (RPN) than some process parameters such as homogenization speed and time, sonication time and amplitude, and temperature. The study established that poloxamer 188 had a more pronounced effect on particle size and drug loading percentage than poloxamer 407 and PS-80 alone.

With the above, we can see that three of the most common CMAs in the evaluation by QbD concerning the stabilization of NP are the concentration of surfactant, the type of surfactant, and the surfactant ratio (Figure 6). Of the three above, the surfactant concentration turns out to be the one that has been most studied in the development of NP, as shown by the review made by Cunha et al. [298]. The authors indicated that the independent variables (CMAs) of lipid(s) and emulsifier(s) concentration produced important effects on the dependent variables (CQAs), mainly in SLN and nanostructured lipid carriers (NLC). However, it has also been observed that the temperature of the stabilizer solution may impact the size and size distribution of dispersion [299], so it can also be considered within the risk analysis to choose the CPPs.

As mentioned, the establishment of CMAs related to NP stabilizers is carried out through risk analysis. The risk analysis to identify and evaluate the type and concentration of the stabilizer as CMAs, during the manufacture of NP, has been carried out mainly through strategies such as Cause and Effect Diagram [300], Failure Mode Effects Analysis (FMEA) [301] and Risk Estimation Matrix (REM) [302]. Despite proving certain differences, all strategies lead to categorizing both variables as critical impacting on CQAs: particle size, zeta potential, entrapment efficiency, PDI, and amount of drug released. However, other tools such as Failure Mode, Effects, and Criticality Analysis (FMECA), Fault Tree Analysis (FTA), Hazard Analysis and Critical Control Points (HACCP), Hazard Operability Analysis (HAZOP), and Preliminary Hazard Analysis (PHA) [209] can also be implemented.

## 8. Drawbacks and Future

As described in previous sections, the diversity of surfactant types is vast, and the possibility of derivatization towards new applications is also possible. It is one of the trends in the search for environmentally friendly surfactants. We detected that although there are different innovations in new structures, there is a predominance of the three mentioned stabilizers, PS, PVA, and poloxamers. However, there is not a complete description of the stabilizer properties in most studies in the case of PVA; while with the PS, some aspects persist about the possible toxic effect in biological models. Concerning poloxamers, the trade names for Lutrol^®^ and Pluronic^®^ make it challenging to distinguish structurally from the type of stabilizer. The new trends outline the lines of research in the search for surfactants of natural origin that allow a high interaction in the interfaces but with high biocompatibility and biodegradation. There may even be modifications of naturally occurring surfactants with synthetic surfactant fragments.

## 9. Conclusions

Non-ionic surfactants offer a wide versatility of applications in the different manufacturing methods of polymeric nanoparticles, highlighting their high biocompatibility and moderate interaction with biological barriers. Reports of significant toxicity are scarce, while the physicochemical parameters of nanoparticles are widely modulated, from particle size to encapsulation capacity. Although some PS, PVA, and poloxamers derivatives exist, the traditional use of primary structures predominates due to the high stability conferred to dispersed systems. The stability-biointeraction balance is necessary to have an adequate performance of the formulation. At the same time, the systematic approach of QbD in the choice of a surfactant is a route that has marked for some years a new and reliable experimental strategy. The outlook in the manufacture of polymeric nanoparticles for biomedical applications seems to indicate that the use of non-ionic surfactants will continue to predominate in the following years due to their ease of application, broad utility, and extensive biosafety background.

## Figures and Tables

**Figure 1 materials-14-03197-f001:**
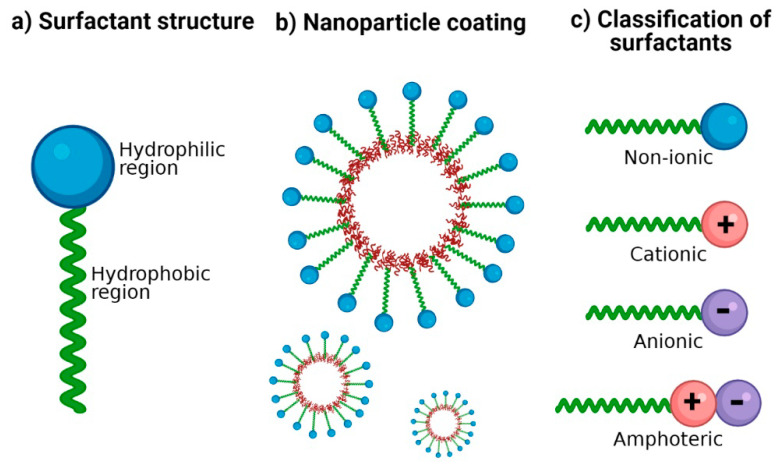
Surfactants for nanoparticle stabilization. (**a**) Classic structure of surfactants: its amphiphilic nature is represented with a hydrophilic region and a hydrophobic region. (**b**) Coating of NP with surfactants: the hydrophobic region possesses an affinity for the nanoparticle surface and the hydrophilic region with an affinity for the aqueous dispersion medium. (**c**) Classification of surfactants according to the ionic charge in its polar group: no charge (non-ionic), positive charge (cationic), negative charge (anionic), and both positive and negative charge (amphoteric).

**Figure 2 materials-14-03197-f002:**
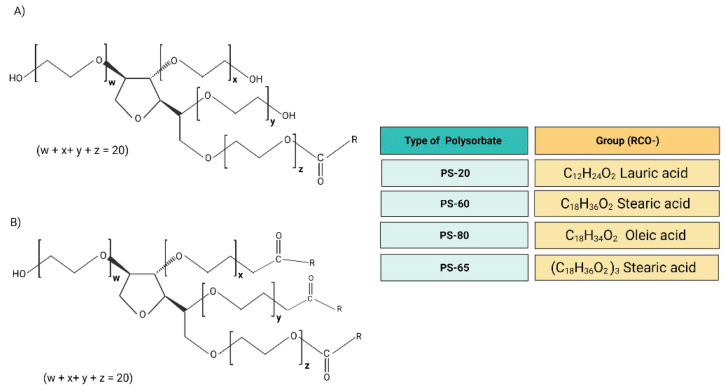
(**A**) Chemical structure of PS-20, PS-60, and PS-80. (**B**) Chemical structure of PS-65. PS-65 presents RCO-groups in x, y, and z, making it a tristearate molecule. The average of the total number of oxyethylene subunits on each polysorbate molecule (w + x + y + z) is 20.

**Figure 3 materials-14-03197-f003:**
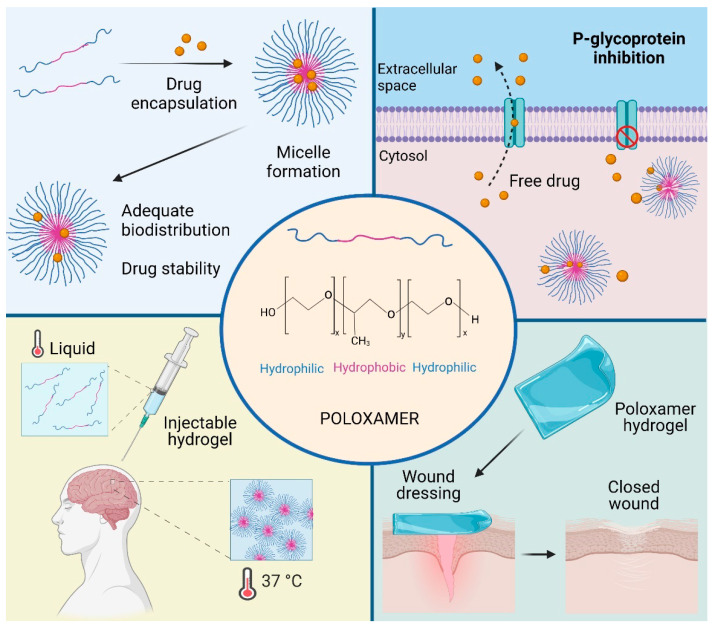
Poloxamer applications. Poloxamers could be used in several fields according to their biological and physicochemical properties. For example, micelles are effectively used as drug carriers. Moreover, they presented the ability to inhibit drug efflux transporters. Poloxamers represent an attractive alternative to tissue engineering, both as injected hydrogel, exploiting their thermosensitivity capacity, and structured hydrogel for wound dressing.

**Figure 4 materials-14-03197-f004:**
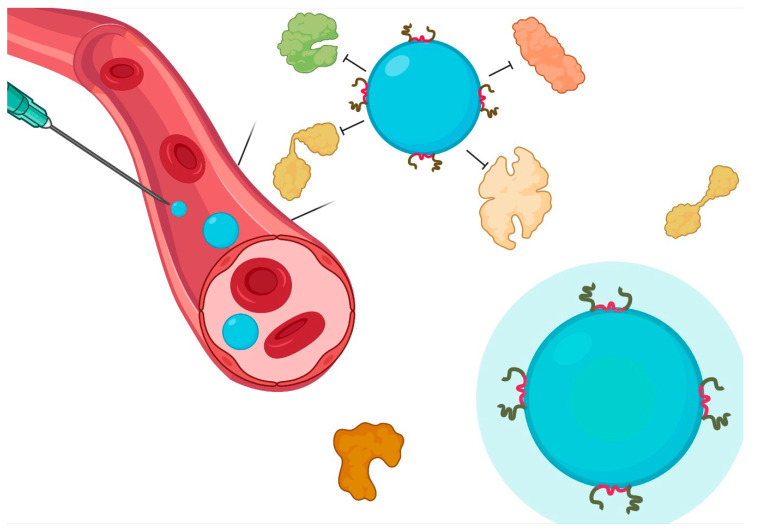
Interaction of non-ionic surfactants in biological systems. Adsorption of PS, PVA, or poloxamer allows a decrease in the deposition of proteins on the surface, increasing the circulation time. Local effects involve an absorption enhancer phenomenon with low inflammatory response.

**Figure 5 materials-14-03197-f005:**
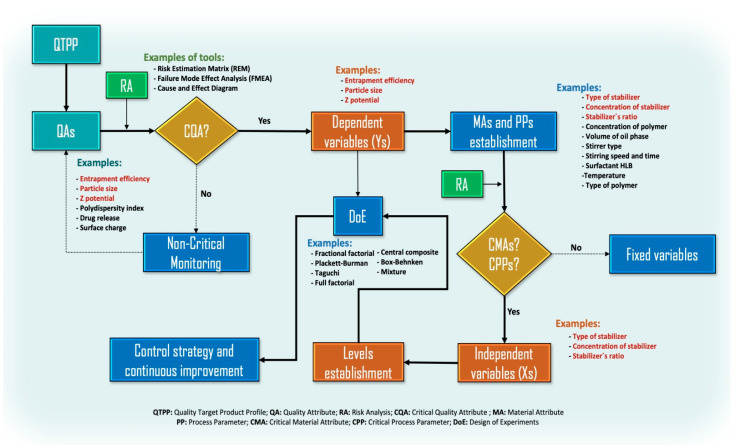
General steps for implementing QbD in the suspension of NP considering the variables related to or affected by surfactants as stabilizers.

**Figure 6 materials-14-03197-f006:**
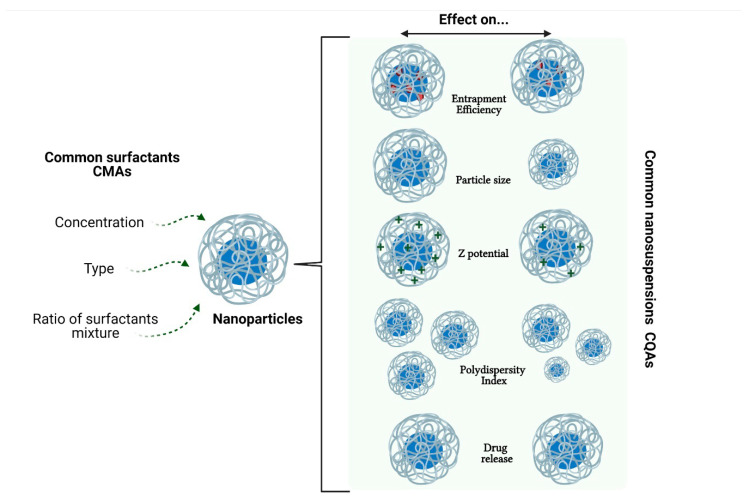
Common CQAs are affected by the most common CMAs of the surfactants.

**Table 1 materials-14-03197-t001:** Examples of surfactants used in pharmaceutical formulations.

Type	Surfactants	References
Anionic	Carboxylates (alkyl carboxylates-fatty acid salts). Sulfates (sodium lauryl sulfate, alkyl ether sulfates). Sulfonates (dioctyl sodium sulfosuccinate, alkyl benzene-sulfonates). Phosphate esters (alkyl aryl ether phosphates, alkyl ether phosphates).	[1,36]
Cationic	Quaternary ammonium (cetrimonium bromide, cetylpyridinium chloride, dimethyldioctadecylammonium chloride). Amine-Based (triethylamine hydrochloride, octenidine dihydrochloride). Pyridinium surfactants (benzethonium chloride)	[32,37]
Non-ionic	Polyol esters (fatty acid esters of sorbitan). Polyoxyethylene esters (polysorbates). Poloxamers (poloxamer 188).	[1,36]
Amphoteric	Phospholipids (phosphatidylcholine or lecithin). Carboxylic Acid/Quaternary Ammonium (cocamidopropyl betaine or amidosulfobetaine-16). Phosphoric Acid/Quaternary Ammonium (hexadecyl phosphocholine). Betaines (alkylamidopropyl betaine).	[32,36,38]

**Table 2 materials-14-03197-t002:** Representative polymeric NP stabilized with polysorbates.

Polymer	Surfactant	NP Size (nm)	PDI	Z-Potential (mV)	Drug	EE or DL	Application	Reference
PLGA	PS-80 1.0%	~120	N.A.	N.A.	Disulfiram	EE = 24%	Anticancer potential in Hep3B cell lines, in vitro model of hepatocellular carcinoma	[60]
PLGA	PS-80 0.1–7.5 mg/mL 10–30 mg/mL	160.5 ± 1.37	0.043 ± 0.0023	−21.53 ± 1.47	N.A.	N.A.	ATR-FTIR method for quantifying the PS-80 adsorbed on the PLGA NP	[61]
PLGA	PS-80 1.0%	~226	0.143	−45.6	Thymoquinone	EE = 69.5 ± 2.97%	Alzheimer’s disease model in albino mice	[14]
PLGA	PS-20 2.0%	155.6 ± 21.8	0.112	N.A.	Rapamycin Piperine	EE ≈ 70%	Polymeric NP for breast cancer treatment	[62]
PLGA PLLA	PS-80 1.0%	50–100	N.A.	−26/−32	N.A.	N.A.	NP crossing BBB model	[63]
PLGA	PS-80 1.5%	~160	0.183	N.A.	Artesunate	DL = 23.67 ± 0.61%	Anticancer activity of artesunate	[64]
PLGA	PS-80 1.0%	77 ± 1	N.A.	−19 ± 0.89	Bacoside-A	EE = 57.11 ± 7.11%	NP for brain targeting of bacoside-A	[65]
PLGA-PS80 16	PS-80 1.0%	~248.1	0.084	−30.9	plasmid DNA	DL = 9.3%	Nanopartcles for gene delivery	[66]
PLGA-Tween 80	PS-80 0.6 mmol	156.5 ± 8.6	0.14	−15.4 ± 1.1	Paclitaxel	DL = 5%	Multidrug resistance lung cancer model	[67]
mPEG-PLGA	PS-80 1%	~145.2	0.133	N.A.	Rhynchophylline	EE = 60%	Neuroprotective effects in an Alzheimer disease model	[68]
PBCA	PS-80 0.0–2.0%	~100	0.018	−2.44	Dalargin	N.A.	Brain targeting of dalargin via oral administration	[69]
Poly (methyl methacrylate-co-methacrylic acid)	PS-60 0.1%	364.03 ± 5.7	N.A.	−29.1 ± 1.9	Gliclazide	EE = 57.46 ± 5.6%	Oral delivery of gliclazide, a hypoglycemic agent	[70]
3-(trimethoxysilyl) propyl methacrylate (TPM)	PS-20 0.01–0.05 mM	~ 15–500	N.A.	−49.0 ± 0.5	N.A.	N.A.	Surfactants influence on spontaneous monodisperse nanoemulsions of TPM	[71]
Polycaprolactone	PS-80	193	0.15	−26.5	Memantine	EE = 80 ± 3%	Alzheimer’s disease approach	[72]
Polycaprolactone	PS-80 200 and 25 (mg/mL)	181–407	0.3–0.5	+11.60 to +29.20	*Rosmarinus officinalis* and *Zataria multiflora* essential oils	EE = 75.8–84.4%	Entrapment of two essential oils against *Tribolium confusum*	[73]
Polycaprolactone	PS-80 2.65 mg/mL 5.3 mg/mL 10.6 mg/mL 21.2 mg/mL	~200	N.A.	−6.73	N.A.	N.A.	Optimization of nanoprecipitation method	[74]
Polycaprolactone	PS-80 0.5%	~170 (uncoated) ~260–360 (chitosan coated)	0.181 (uncoated) 0.345 (chitosan coated)	−12.91 (uncoated) +31.73 (chitosan coated)	Paliperidone	EE ≈ 60%	Influence of PCL/drug ratio, stabilizer type, and high molecular weight of chitosan coating	[75]
Polystyrene	PS-20 0.1%	20–200	N.A.	N.A.	N.A.	N.A.	NP distribution after periocular administration	[76]
Poly (sebacicanhydride) (PSA)	PS-60 PS-20	200–160	N.A.	N.A.	N.A.	N.A.	Degradation of PSA NP	[77]
Polyhydroxybutyrate (PHB)	PS-80 1% *v/v*	146 ± 30	N.A.	−26	Carvacrol	DL = 12.5%	Preparation and characterization of PHB NP by nanoprecipitation and dialysis methods	[78]
Chitosan-folate conjugated	PS-80 0.5% *v/v*	111.8 ± 4.11	0.50 ± 0.21	N.A.	Doxorubicin and curcumin analog	N.A.	Concentration of PS-80 decreased the size of NP	[79]
Chitosan and chondroitin sulfate	PS-80 15 mg	~234	0.2	+30.0	Artemether	EE = 83 ± 0.28%	Transdermal antimalarial drug delivery system	[80]
Chitosan	PS-80 1.25%	208 ± 0.01	N.A.	−32.56 ± 0.03	Imatinib	EE = 68.52 ± 0.01%	Colorectal cancer targeting application	[81]
Sodium Alginate	PS-80	~383	0.2	200	Curcumin	EE = 95%	Bioavailability in healthy human volunteers	[82]

Abbreviations: N.A. = Not Available; EE = Entrapment efficiency; DL = Drug Loading; PDI = Polidispersity Index; PS = Polysorbate; PDI = Polidispersity Index; PLLA = Poly(l-lactic acid); PLGA = Poly(lactic-co-glycolic acid).

**Table 3 materials-14-03197-t003:** Nanoparticulate formulations stabilized with PVA.

Polymer	Surfactant	NP Size (nm)	PDI	Z-Potential (mV)	Drug	EE or DL	Application	Reference
PLGA	2%	215.3 ± 23	0.071	−10.3 ± 2.3	Chlorogenic acid	DL = 2.25 ± 0.21%	As a promotor of type 17 collagen production	[165]
PLGA	1–2% *w/v* and 1.5% *w/v*	150 ± 10.4	0.081 ± 0.030	17.7	IFN-beta-1a	EE = 96.2%	To diminish symptoms of relapsing-remitting multiple sclerosis	[166]
PLGA	0.1%	213.8 ± 34.99	0.232 ± 0.021	−52.6 ± 9.483	Curcumin and ovalbumin	EE = 30% Cur; 16% Ova	Use as sublingual immunotherapy (SLIT) in a mouse model of rhinitis allergic	[167]
PLGA	2%	172.6 ± 6.20 to 271.9 ± 18.2	0.070 ± 0.02 to 0.301 ± 0.03	N.A.	Ketoconazole	EE = 94.99% ± 3.45 to 97.53% ± 2.33	Treatment against *Candida albicans*	[168]
PLGA	2%	198.6 ± 5.4 (before freeze-drying) 299.8 ± 2.2 (after freeze-drying)	0.160 ± 0.033 (before) 0.412 ± 0.028 (after)	−20.8 ± 1.4 (before) −16.6 ± 1.1 (after)	Bevacizumab	DL = 1.62 ± 0.01% (before freeze-drying)	Antiangiogenic therapy	[169]
PLGA	0.3% (*w/v*)	140	0.463	N.A.	Farnesol	N.A.	Antibiofilm activity, against *Candida albicans*	[170]
PLGA	0.5% to 5%	127 ± 0.90 to 289 ± 1.56	0.191 ± 2.66 to 0.259 ± 2.67	−30.43 to −30.89	Bicalutamide	N.A.	For the treatment of prostate cancer	[171]
PLGA	0.3% (*w/w*), 1.0% (*w/w*), and 3.0% (*w/w*)	121 to 259	0.05 to 0.20	−27 to −34.4	FLAP/PGES-1 Inhibitor BRP-187	DL = 0.5 to 7.29%	A promising drug candidate due to its improved anti-inflammatory efficacy with potentially reduced side-effects in comparison with NSAIDs	[172]
PLGA	1%	183.7 ± 72.21	N.A.	−41.1 ± 4.81 mV	p66shc siRNA	EE = 32.3%	To ameliorate neuropathic pain following spinal nerve ligation	[173]
PLGA	1.0% (*w/v*)	110.0 ± 41.0	N.A.	N.A.	17 beta-estradiol	N.A.	To improve low bone mineral density of cancellous bone caused by osteoporosis	[174]
PLGA	4%	211 ± 74	N.A.	−14.2 ± 0.8	Thioridazine	DL = 26.5%	To reduce toxicity against mycobacterial infection in zebrafish	[175]
PLGA	1%	198 ± 0	0.16 ± 0.01	−78 ± 1	Combretastatin A4	EE = 32 ± 3% DL = 0.41 ± 0.02%	To improve physicochemical characteristics of combretastatin A4, a natural potent tubulin polymerization inhibitor	[176]
PLGA	1%	119 ± 9 to 206 ± 27	0.220 to 0.401	−4.38 to −5.24	Curcumin	EE = 77.81 to 92.64% DL = 7.86 to 10.53%	Toxicity on human glioblastoma U87MG cells	[177]
PLGA	0.5% *w/v*	186.6	0.108	−28.8	Recombinant ArgF	EE = 76% DL = 2.6%	For potential use for the prevention of *Mycobacterium bovis* infection	[178]
PLGA	5.21 mg/mL	202.8 ± 2.64	0.17 ± 0.016	N.A.	Resveratrol	EE = 89.32 ± 3.51%	For prostate cancer cells	[179]
PLGA	1%	225.9	0.257	−10.9	Curcumin and Niclosamide	EE = 58.31% Cur and 84.8% Nic DL = 2.92% Cur and 4.24% Nic	To improve therapeutic effect on breast cancer cells	[180]
PLGA	0.5% *w/w*	496 ± 8.5	0.607	−18.41 ± 3.14	Rivaroxaban	EE = 87.9 ± 8.6% DL = 9.5 ± 1.6%	Anticoagulant medication to prevent blood clots	[181]
PLGA	1% *w/v*	110 ± 1	0.117 ± 0.003	−1.29 ± 0.35	Doxorubicin	EE = 80%	For chemotherapy of glioblastoma	[182]
PLGA	1% *w/v*	527 ± 50.21	0.26	N.A.	Olmesartan medoxomil	EE = 78.65 ± 4.31%	To increase the bioavailability of the drug to treat hypertension	[183]
PLGA	1% *w/v*	180 ± 8	0.04	−8.59 ± 0.20	Paclitaxel and methotrexate	EE = 70% MTX and 88% PTX DL = 4% MTX and 5% PTX	Treatment against glioblastoma	[184]
PLGA	3% *w/v*	152.8 ± 2.65	0.187 ± 0.024	−30.9 ± 1.67	Lipophosphoglycan molecule (LPG) and leishmania antigen (ALA)	EE = 14% ALA, 28% LPG DL = 28% ALA, 12% LPG	For a potential nanovaccine to prevent leishmaniasis	[185]
PLGA	0.5% *w/v*	114.7 to 124.8	0.113 to 0.147	N.A.	Diclofenac sodium	EE = 41.4% to 77.8%	For inflammatory diseases	[186]
PLGA	3% *w/v*	252.6 ± 2.854	0.209 ± 0.008	−23.7 ± 1.36	Rutin	EE = 81 ± 5%	As a candidate for further multidisciplinary studies (support blood circulation, allergies, viruses, etc.)	[187]
PLGA	4% *w/v*	182.2 ± 11.40	0.147 ± 0.01	N.A.	Doxorubicin	EE = 75.3% DL = 4.9%	To arrest glioblastoma growth via intranasal delivery	[188]
PLGA	3%	191.92 ± 2.3 to 273.70 ± 1.9	0.070 ± 0.014 to 0.237 ± 0.030	−6.87 ± 0.1 to −11.5 ± 0.4	Dexamethasone	EE = 94.39 ± 2.70% to 95.02 ± 2.98% DL = 3.27 ± 0.58% to 5.43 ± 0.63%	Potential treatment of oral precancerous lesions	[189]
PLGA	2% *w/v*	229.5 ± 38.4 to 379.2 ± 21.6	0.36 ± 0.02 to 0.73 ± 0.13	−1.2 ± 1.1 to −3.9 ± 0.5	Ethanolic Extract of Propolis	EE = 89.90 ± 0.8% to 92.1 ± 0.5% DL = 28.6 ± 1.1% to 56.7 ± 3.4%	As a treatment against *Candida albicans*	[190]
PLGA	1%	97.36 ± 2.01	0.263 ± 0.004	−17.98 ± 1.09	Thymoquinone	EE = 82.49 ± 2.38% Dl = 5.09 ± 0.13%	For the treatment of epilepsy	[191]
PLGA	2% *w/v*	200 ± 05	0.05 ± 0.02	N.A.	Budesonide	EE = 85 ± 3.5%	To target the inflammation of mucosa	[192]
PLGA	1.0% *w/v*	277	0.18	−16	Cymbopog citratus essential oil	EE = 73%	As a vehicle for this essential oil with anti-inflammatory, antifungal, sedative, antibacterial, antiviral and anticarcinogenic properties	[193]
PLGA	1% *w/v*	118 to 279	0.103–0.581	N.A.	Quercetin	EE = 73.55 ± 2.11% to 86.48 ± 1.67%	Potential vehicle for the antioxidant quercetin	[194]
PLGA	5% *w/v*	105 ± 3	N.A.	−36 ± 5	Surfactant Protein D (SP-D)	EE = 59 ± 4%	As a potential treatment for respiratory distress syndrome in preterm infants	[195]
PLGA	2% *w/v*	154 ± 4.56	0.29	−18.4	Ursolic Acid	EE = 40 ± 3.24% DL = 4 ± 1.12%	Potential vehicle to deliver the drug against different bearing cell lines	[196]
PLGA	0.5–1.5% *w/v*	200	N.A.	−17.5	Zaleplon	DL = 5%	For treatment of insomnia	[197]
PLGA	1% *w/w*	244.3 ± 4 to 262.8 ± 7	N.A.	−8.8 ± 0.8 to −17.4 ± 1.0	Quercetin	EE = 96.2 to 97.8%	To treat foodborne pathogens	[198]
PLGA	1% (*w/v*)	211 ± 3	0.211 ± 0.009	N.A.	Clofazimine	DL = 12 ± 1%	To decrease toxicity of the antimicrobial drug	[199]
PLGA	1.5% (*w/v*)	268 ± 2.7	0.110 ± 0.026	N.A.	Atrazine	EE = 31.6 to 50.5%	Potential herbicide release system for agriculture	[200]
PLGA	1%	192.6 ± 3.5	0.234 ± 0.008	−32.4	Atenolol	EE = 71.65 ± 1.8%	Drug carrier of a β-blocker for cardiovascular disorders	[201]
PLGA	2% (*w/v*)	294 ± 15	0.26 ± 0.02	−20.4 ± 2.5	Insulin	DL = 12.1 ± 0.6%	To optimize the PLGA formulation and lyophilization	[202]
PLGA	2.5%	184 ± 7	0.19 ± 0.01	−21.7	Rhodamine-B	EE = 40 ± 2.94%	Potential probes for the drug delivery in cardiac myocytes	[203]
PLGA	2% *w/v*	133.3 ± 10.4	0.087 ± 0.009	−16.1 ± 4.5	Trastuzumab	EE = 42.8 ± 1.6%	Potential vehicle for therapeutic antibodies to avoid current limitations	[204]
PLGA	0.5% *w/v*	281.9 to 307.3	0.317 to 0.451	−32.8 ± 1.6 to −43.4 ± 2.6	Apremilast	EE = 39.5 ±1.1% to 61.1 ± 1.9% DL = 1.3 ± 0.1% to 1.9 ± 0.1%	For the treatment of psoriasis or psoriatic arthritis	[205]
PLGA	3% *w/v*	150 ± 7	0.16 ± 0.05	−23.8 ± 0.8	Rifapentine	EE = 85 ± 8%	For a treatment against tuberculosis	[206]
PLGA	2.5% (*w/v*) and 0.25% (*w/v*)	157.7	0.071	−35.1	Indocyanine Green and Resiquimod	EE = 65.61 ± 2.09% ICG and 8.363 ± 0.325% R848	For prostate cancer treatment	[207]
PLGA	1%	226.6 ± 44.4	0.039 ± 0.013	−0.144	Ketotifen Fumarate	EE = 89.3 ± 3.3%	Vitamin D binding protein	[208]
PLGA	1% (*w/v*)	120 ± 1	0.104 ± 0.011	−11.6 ± 0.8	Doxorubicin	EE = ~80%	For the delivery of the drug into U87 human glioblastoma cells	[209]
PLGA	0.5% *w/v*	210.0 ± 4.8 to 317.5 ± 4.7	0.190 ± 0.39 to 0.394 ± 0.53	−8.3 ± 2.1 to −19.3 ± 0.2	Dexamethasone	EE = 10.4 ± 2.6% to 64.9 ± 0.6% DL = 0.67 ± 0.2% to 7.17 ± 3.2%	For treatment of oral mucositis	[210]
PLGA	1% *w/v*	167.6 ± 0.37	0.118	−16.17 ± 0.53	Loteprednol etabonate	EE = 96.31 ± 1.68%	For the delivery of the drug into the cornea	[211]
PLGA	1% (*w/v*)	164.6	0.203	−17.6	Doxorubicin and Sorafenib	EE = 74% Dox, 67% Sor	For a cancer treatment using nanotherapeutics	[212]
PCL	3.0%	188.5 ± 1.7	0.160 ± 0.022	−15.03 ± 2.83	Lapazine	EE = 35.82 ± 1.47% DL = 54.71%	For antitubercular treatment	[213]
PCL	N.A.	202 ± 24 to 389 ± 37	0.08 to 0.164	−4.92 ± 0.88 to −8.29 ± 1.04	Nigella sativa oil	EE = 71.6 to 98.6%	For leishmaniasis treatment	[214]
PCL	2.0% *w/v*	311.6 ± 4.7	0.21 ± 0.03	−16.3 ± 3.7	Carboplatin	EE = 27.95 ± 4.21%	Intended to use for intranasal administration to improve brain delivery	[215]
PCL	2% to 3%	275.23 ± 4.56 to 452.30 ± 9.02	0.09 to 0.35	−4.41 ± 1.21 to −14.77 ± 4.42	5-fluorouracil	EE = 94.53 ± 0.23% to 96.82 ± 0.46%	For colon cancer treatment	[216]

Abbreviations: N.A. = Not Available; EE = Entrapment efficiency; DL = Drug Loading; PDI = Polidispersity Index; PLGA = Poly(lactic-co-glycolic acid); NSAIDs = Non-steroidal anti-inflammatory drugs.

**Table 4 materials-14-03197-t004:** Nanoparticulate formulations stabilized with poloxamers.

Polymer	Surfactant	NP Size (nm)	PDI	Z-Potential (mV)	Drug	EE or DL	Application	Reference
PCL	F-127, 2%	167	0.188	~0	Amphotericin	EE = 85%	Increase the solubility of the drug as treatment for *Leishmania* infections	[247]
Chitosan	F-127, 15%	146	N.A	5.09	Curcumin	EE = 61.7%	Development of effective delivery system with few side-effects	[250]
PCL	F-108, 50% F-127, 50% F-68, 50%	182 184.7 698.4	0.2 0.28 0.88	–11.7 –1.6 –6.03	N.A	N.A	Evaluation of the effect of different surfactants	[251]
PLGA	F-68, 0.88%	217.6 ± 8.6	0.171	–23.35 ± 1.17	Docetaxel	EE = 88%	Development of a delivery system for breast cancer chemotherapy	[252]
PLGA	F-68, 0.5%	160–170	0.051 ± 0.012	−20.5 ± 0.069	N.A	N.A	Evaluation of the effect of poloxamer as surfactant	[253]
PCL/F-68	PVA, 0.05%	201.7 ± 10.1	0.096	–12.50 ± 0.86	Docetaxel	EE = 69.1% DL = 10%	Evaluation of the increased level of uptake NP due to F-68 presence	[254]
PCL	F-68, 2%	149.9 ± 2.2	0.087 ± 0.05	N.A	Curcumin	EE = 96 ± 0.95% DL = 4.9 ± 0.7%	Development of a potential alternative treatment for neuronal diseases based on curcumin	[244]
PS	L61, F-68, F-108, L121, F-127	97 ± 1, 105 ± 1, 110, 100 ± 2, 108 ± 1	0.01, 0.03, 0.02, 0.02, 0.02	–42 ± 1, –26 ± 2, –14 ± 2, –32 ± 1, –18 ± 2,	N.A	N.A	Analysis of polymer NP modified with different types of poloxamers	[255]
Chitosan	F-68 0.5%	252.80 ± 7.46	0.40 ± 0.03	17.50 ± 0.93	Doxorubicin	EE = 61.3 ± 2.28%	Fabrication of DOX-loaded pH-responsive NP for chemotherapy	[256]
Silk sericin	F-127 (1:5) F-87 (1:5)	61.9 ± 5.36 103 ± 1.0 nm	0.21 0.18	N.A	Inulin Paclitaxel	EE = 65 ± 10%	Development of silk sericin NP in the presence of poloxamer for successful delivery of both hydrophobic and hydrophilic drugs	[257]
PLGA/F-68	PVA 1%	179.4 ± 11.2	0.309± 0.08	−22.7 ± 5.7	Paclitaxel	EE = 65 ± 8.3%	Development of novel PLGA:poloxamer blend NP for intravenous administration of paclitaxel	[258]
PLGA/F-68/F-127	PVA, 1%	160 ± 31	0.671 ± 0.03	18.7 ± 1.3	Curcumin	EE = 90 ± 2.1%	Obtention of PLGA/poloxamer blend NP and evaluation of their interaction with serum proteins and its internalization ability	[259]
PLGA-Chitosan	F-68, 1%	150.7 ± 1.8	0.16 ± 0.03	25.1 ± 1.6	miR-34a	EE = 49 ± 2.1%	Anticancer treatment of multiple myeloma	[260]
PLGA-Chitosan	F-68, 1%	∼130	N.A	30	Anti-hTERT siRNA	N.A	Block the growth of anaplastic thyroid cancer xenograft	[261]
Chitosan	F-68, 10–50%	~122	N.A	23.63	Doxazosin mesylate	EE = 99.9% DL = 8.5%	Control release and enhancing the bioavailability of doxazosin mesylate	[262]
PLGA	F-68, 1%	~94	0.091 ± 0.010	–0.3	Doxorubicin	EE = 92%	Treatment of glioblastoma	[263]
PLGA/Chitosan	F-68, 5%	~134.4	N.A	43.1	Insulin	EE = 52.8% DL = 1.3%	Characterization of bioadhesive NP for oral administration	[264]
PLGA	P-85	156.7 ± 3.9	0.21 ± 0.04	–45.7 ± 2.9	Doxorubicin	EE = 85.2 ± 4.1% DL = 7.3 ± 1.2%	Treatment of leukemia	[265]
PLGA	P-85/PVA	180.26 ± 5.60	0.184	−17.47 ± 2.67	Docetaxel	EE = 82.7% DL = 10%	Breast cancer treatment	[266]
Chitosan-γPGA	F-127, 0.25–1%	193.1 ± 8.9	0.29 ± 0.02	20.6 ± 2.4	Curcumin	EE = 52.8 ± 4.7%	Wound regeneration	[267]
PCL	F-127, 0.06%	∼123.5	N.A	–29.6	Chloramphenicol	EE = 98.3%	For treatment of MRSA-infected burn wounds	[268]
Folated F127/PLGA	F-127	107.6 ± 4.25	0.308 ± 0.01	N.A	Paclitaxel	EE = 3.4%	Prolongation of the circulation time of paclitaxel	[269]
F-127	F-127, 0.02%	9.70 ± 0.31	0.195 ± 0.029	–27.01 ± 0.20	Berberine	EE = 87.6 ± 1.52%	Improve permeability and retention in the skin	[270]
PLGA	F-127 and F-108, 0.2%	~115	<0.1	–11.3	N.A	N.A	Functionalization of polymeric NP	[271]
F-127	F-127, 1.2%	70 ± 2.4	0.12	N.A	Gossypol	EE = 91.2 ± 3.1% DL = 9.1 ± 0.42%	Cancer drug release study	[272]
Trimethyl chitosan	F-127, 0.1%	~160	0.140	+20.1	Methotrexate	EE = 93.6% DL = 8.95%	Effective delivery of methotrexate in osteosarcoma	[273]
PLGA	F-127, 1%	159.0 ± 3.0	0.099 ± 0.042	–15.4 ± 0.7	Rose Bengal	DL = 0.82 ± 0.27%	Evaluation of the effect of the nanoparticle delivery system on the biodistribution of the drug	[274]
PLGA/F-68	N.A.	154	0.118	–25.2 ± 1.1	PDGF-BB	EE = 87 ± 2%	Development of injectable controlled release device based on polymeric NP for the delivery of growth factors.	[275]

Abbreviations: N.A. = Not Available; EE = Entrapment efficiency; DL = Drug Loading; PBCA = Poly(butyl cyanoacrylate); PCL = Poly (e-Caprolactone); PDGF-BB = platelet derived growth factor; PDI = Polidispersity Index; PLGA = Poly(lactic-co-glycolic acid); PS = Poly(styrene; F-68 = poloxamer 188; F-87 = poloxamer 238; F-108 = poloxamer 338; F-127 = poloxamer 407; L61 = poloxamer 181; L121 = poloxamer 401; P-85 = poloxamer 235.

## Data Availability

Data is contained within the article.

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
