# Peer review of "Non-Ionic Surfactants for Stabilization of Polymeric Nanoparticles for Biomedical Uses"

_materials, 2021, doi:10.3390/ma14123197_

Round 1

Reviewer 1 Report

It is a well written and comprehensive review by Cortes and coworkers about non-ionic surfactants for stabilization of polymeric nanoparticles for biomedical uses. I recommend it for publication in Materials after the following points are addressed.

1. Line 26, ‘polysorbates, PVA’ should be changed to ‘polysorbates (PS), poly(vinyl alcohol) (PVA)’.

2. Line 136-140, this comment on the pH is not accurate. Zwitterionic surfactants could be zero charge at a wide range of pH. It would not be easily positively charged at acid condition, or negatively charged at basic condition. Several recent studies (doi.org/10.1021/acs.langmuir.1c00178; doi.org/10.1021/acsnano.0c01559; 10.1039/D0SM00399A) related to zwitterionic surfactants should be included.

3. For the readers, the authors could add several figures describing some typical studies.

4. Line 215-223, they must add more discussion about different PS (mainly one figure about chemical structures of different PS). Otherwise, the readers will be confused by these numbers (such as PS-20, PS-60, PS-65).

Author Response

Dear reviewer. Thank you for your valuable comments, your observations allowed us to strengthen our manuscript.

Reviewer 1

It is a well written and comprehensive review by Cortes and coworkers about non-ionic surfactants for stabilization of polymeric nanoparticles for biomedical uses. I recommend it for publication in Materials after the following points are addressed.

  1. Line 26, ‘polysorbates, PVA’ should be changed to ‘polysorbates (PS), poly(vinyl alcohol) (PVA)’.

- We made the change suggested for the reviewer.

  1. Line 136-140, this comment on the pH is not accurate. Zwitterionic surfactants could be zero charge at a wide range of pH. It would not be easily positively charged at acid condition, or negatively charged at basic condition. Several recent studies (doi.org/10.1021/acs.langmuir.1c00178; doi.org/10.1021/acsnano.0c01559; 10.1039/D0SM00399A) related to zwitterionic surfactants should be included.
  • We thank this valuable observation and apologize for the error in the expression of the concepts. We corrected that paragraph and included four new references.

  1. For the readers, the authors could add several figures describing some typical studies.
  • We thank this timely suggestion. We have now added a figure to exemplify some poloxamer applications (Figure 3).

  1. Line 215-223, they must add more discussion about different PS (mainly one figure about chemical structures of different PS). Otherwise, the readers will be confused by these numbers (such as PS-20, PS-60, PS-65).
  • According to this opportune observation, we modified that paragraph and added a new figure to show the chemical structures of PS-20, PS-60, PS-65, and PS-80.

Reviewer 2 Report

Introduction

Page 1 rows 37-38  Regarding the classical definition of surfactants cited by the authors in the beginning of Introduction paragraph that they are “are amphipathic molecules with two defined segments, a hydrophilic and hydrophobic section” I consider that is an incomplete and very narrow one since for example the poloxamers have one (middle) hydrophobic section and two symmetric separate hydrophilic sections/segments, situation which do not correspond to the classical definition of surfactant. This narrow definition is even more inappropriate for PVA, where one can found alternating 1,3 diols (majority) and 1,2 diols (minority) segments,  so methylene hydrophobic groups are intercalated between –CHOH- or even -CHOH-CHOH-  hydrophilic groups , so there are not only two but many more hydrophilic and hydrophobic ALTERNATING segments. So the authors are invited to expand this narrow definition in order tobe accurate  also for poloxamers and PVA cases May be a more general definition for surfactants can be foundin the literature

Page 1 rows 49-52 The phrase “The initial globule size of emulsified dispersed systems is greater than  the colloidal particle size at the end of the manufacturing process, combined with a high  surface free energy and, therefore, the tendency of nanoparticles (NP) to flocculate and  coagulate [3].” should be clarified and changed for example as “The initial globule size of emulsified dispersed systems is greater than  the colloidal particle size at the end of the manufacturing process, combined with the presence of a high  surface free energy and, therefore, the tendency of resulted nanoparticles (NP) to flocculate and  coagulate can be observed [3].”

Page 1 row 67  The phrase “This work is a tribute to the principal non-surfactants utilized in the manufacture of…’’   should be corrected as   “This work is a tribute to the principal non-ionic surfactants utilized in the manufacture of…”

Page 3 rows 103-104  The phrase “these explain the  position and approximate inclination angle surfactants and their physicochemical implications in the surface coating [9,13].“ should be corrected as “these explain the  position and approximate inclination angle of adsorded surfactant molecules and their physicochemical implications in the surface coating [9,13].”

Page 4 row 135  The phrase “However, they develop ionic behaviors when reacting with the pH of the emulsion;”should be corrected as “However, they develop ionic behaviors in response to the  pH of the emulsion;”

Page 5 – At the paragraph 2.4 New surfactants, the authors should also introduce the PEG-ylated amides class, an important example here being ALC-0159  H(OCH2CH2)45-CH2-CO-N(nC14H29)2  used by Pfizer-BionNTech in their mARN (messenger Ribonucleic Acid) -based COVID19 vaccine formulation. Also , in the same vaccine formulation, for the mARN stabilization, that producer used a non-ionic yet ionisable hidroxyaminodiester surfactant molecule ALC-0315  HO(CH2)4N[(CH2)6OCOCH(nC6H13)(nC8H17)]2

At page 5 the authors should introduce also other natural surfactants  beside the listed zwitterionic phospholipids such as ceramides, cerebrosides and ganglioside, these three types belonging to  the non-ionic surfactants class. Ganglioside are related with the alkylpolyglycosides class mentioned by authors at the rows 169-171. Some of them can have also bioactive role [see for example H. Devalapally et al. Paclitaxel and ceramide co-administration in biodegradable polymeric nanoparticulate delivery system to overcome drug resistance in ovarian cancerInt. J. Cancer: 121, 1830–1838 (2007)]

Row 184 –Here the PLGA  polymer is mentioned for the first time in manuscript , so the PLGA abbreviation the should be defined as poly(lactic-co-glycolic acid)

Row 268 – Here the CMC abbreviation should be defined as Critical Micelle Concentration

Row 256 and also Table 2 the abbreviation PDI should be defined

Rows 273-274  The phrase “Similarly, the presence of the PS80 on the nanoparticle surface improves the cross of the blood-brain barrier (BBB).’’ should be corrected as   “Similarly, the presence of the PS80 on the nanoparticle surface improves the crossing of the blood-brain barrier (BBB).’’

Row 277  The phrase ‘’in consequence, the NP treated with PS-80 enhance the reach of the drug to the brain [48,49].’’ should be modified as ‘’in consequence, the NP treated with PS-80 enhance the chance of the drug to reach the brain [48,49].’’

In the Table 2 at pag 10 the authors introduced the PLLA  abbreviation which should be defined as poly(l-lactic acid)

In the Table 2 at reference [45] the abbreviation pDNA should be defined

Row 344  The phrase ‘’generally by hydrolysis (methanolysis) of polyvinyl-acetate (PVAc) “ should be clarified as ‘’generally by hydrolysis or methanol transesterification (methanolysis) of polyvinyl-acetate (PVAc) “

Row 346  The claim regarding PVA  ‘’that is resistant to many solvents‘’ should be detailed with examples and clases of solvents

Other acronyms that should be defined: row 402 CS-DS-Dox; rows 404, 405 PAA from PAA-b-PVA; row 407 HEC from PVA-HEC; row 416 PMMA from CS-g-PMMA:PVA-g-PMMA; also the meaning of letter g should be introduced (I presume that indicates the grafting of polymeric chains)

Row 429 The phrase "which could be beneficial or unfortunate. " should be corrected as "which could be beneficial or harmful. "

Rows 472-475 The following claims " Interestingly, several reports demonstrated that poloxamers could 472 decrease the cell membrane microviscosity (membrane fluidization) due to hydrophobic PPO chains. Furthermore, it has been reported that poloxamers could decrease multidrug resistance and induce a dramatic reduction in ATP levels in cancer and barrier cells and  inhibit drug efflux transporters. " should be supported by adequate references

Row 584  The chemical composition/name  of Brij35 (C12H25(OCH2CH2)23OH and SDS (Sodium dodecyl sulphate) should be introduced

Row 617 The abbreviation SE from Laurate SE should be defined

Rows 618-619 The abbreviation    cremophor EL     should be defined

Row 629    Abbreviation AUC should be defined

Row 638    Abbreviation ICH should be defined

A point which is must be introduced in this review is the immunogenic potential of these surfactants, especially those that contain polyethoxylated chains - see for example the review article [Nicola d’Avanzo, Christian Celia, Antonella Barone, Maria Carafa, Luisa Di Marzio, Hélder A. Santos, Massimo Fresta Immunogenicity of Polyethylene Glycol Based Nanomedicines: Mechanisms, Clinical Implications and Systematic Approach,  Adv. Therap. 2020, 1900170]

Author Response

Dear reviewer. Thank you for your valuable comments, your observations allowed us to strengthen our manuscript.

Reviewer 2

Introduction

Page 1 rows 37-38  Regarding the classical definition of surfactants cited by the authors in the beginning of Introduction paragraph that they are “are amphipathic molecules with two defined segments, a hydrophilic and hydrophobic section” I consider that is an incomplete and very narrow one since for example the poloxamers have one (middle) hydrophobic section and two symmetric separate hydrophilic sections/segments, situation which do not correspond to the classical definition of surfactant. This narrow definition is even more inappropriate for PVA, where one can found alternating 1,3 diols (majority) and 1,2 diols (minority) segments,  so methylene hydrophobic groups are intercalated between –CHOH- or even -CHOH-CHOH-  hydrophilic groups , so there are not only two but many more hydrophilic and hydrophobic ALTERNATING segments. So the authors are invited to expand this narrow definition in order to be accurate  also for poloxamers and PVA cases may be a more general definition for surfactants can be found in the literature

  • We agree with this valuable observation and apologize for the error in the expression of the concepts. We now show a general description that includes the three surfactants in the manuscript.

Page 1 rows 49-52 The phrase “The initial globule size of emulsified dispersed systems is greater than  the colloidal particle size at the end of the manufacturing process, combined with a high  surface free energy and, therefore, the tendency of nanoparticles (NP) to flocculate and  coagulate [3].” should be clarified and changed for example as “The initial globule size of emulsified dispersed systems is greater than  the colloidal particle size at the end of the manufacturing process, combined with the presence of a high  surface free energy and, therefore, the tendency of resulted nanoparticles (NP) to flocculate and  coagulate can be observed [3].”

- In order to clarify the sentence, we made the change suggested for the reviewer.

Page 1 row 67  The phrase “This work is a tribute to the principal non-surfactants utilized in the manufacture of…’’   should be corrected as   “This work is a tribute to the principal non-ionic surfactants utilized in the manufacture of…”

- In order to clarify the sentence, we made the change suggested for the reviewer.

Page 3 rows 103-104  The phrase “these explain the  position and approximate inclination angle surfactants and their physicochemical implications in the surface coating [9,13].“ should be corrected as “these explain the  position and approximate inclination angle of adsorded surfactant molecules and their physicochemical implications in the surface coating [9,13].”

  • We made the change suggested for the reviewer to clarify the sentence.

Page 4 row 135  The phrase “However, they develop ionic behaviors when reacting with the pH of the emulsion;”should be corrected as “However, they develop ionic behaviors in response to the pH of the emulsion;”

  • We made the change suggested for the reviewer to clarify the sentence.

Page 5 – At the paragraph 2.4 New surfactants, the authors should also introduce the PEG-ylated amides class, an important example here being ALC-0159  H(OCH2CH2)45-CH2-CO-N(nC14H29)2  used by Pfizer-BionNTech in their mARN (messenger Ribonucleic Acid) -based COVID19 vaccine formulation. Also , in the same vaccine formulation, for the mARN stabilization, that producer used a non-ionic yet ionisable hidroxyaminodiester surfactant molecule ALC-0315  HO(CH2)4N[(CH2)6OCOCH(nC6H13)(nC8H17)]2

  • We thank this valuable suggestion. We added more examples that included some PEG-ylated amides and glycolipids.

At page 5 the authors should introduce also other natural surfactants  beside the listed zwitterionic phospholipids such as ceramides, cerebrosides and ganglioside, these three types belonging to the non-ionic surfactants class. Ganglioside are related with the alkylpolyglycosides class mentioned by authors at the rows 169-171. Some of them can have also bioactive role [see for example H. Devalapally et al. Paclitaxel and ceramide co-administration in biodegradable polymeric nanoparticulate delivery system to overcome drug resistance in ovarian cancerInt. J. Cancer: 121, 1830–1838 (2007)]

  • We thank this valuable suggestion. We added more examples that included some glycolipids and ceramides.

Row 184 –Here the PLGA  polymer is mentioned for the first time in manuscript , so the PLGA abbreviation the should be defined as poly(lactic-co-glycolic acid)

  • We thank this observation. We have now defined the acronym.

Row 268 – Here the CMC abbreviation should be defined as Critical Micelle Concentration

  • Thanks for the observation. We defined this acronym at its first mention (in: 3.1 Physicochemical properties)

Row 256 and also Table 2 the abbreviation PDI should be defined

  • Thanks for the observation. We defined this acronym at its first mention and Table 2.

Rows 273-274  The phrase “Similarly, the presence of the PS80 on the nanoparticle surface improves the cross of the blood-brain barrier (BBB).’’ should be corrected as   “Similarly, the presence of the PS80 on the nanoparticle surface improves the crossing of the blood-brain barrier (BBB).’’

  • We made the change suggested for the reviewer to clarify the sentence.

Row 277  The phrase ‘’in consequence, the NP treated with PS-80 enhance the reach of the drug to the brain [48,49].’’ should be modified as ‘’in consequence, the NP treated with PS-80 enhance the chance of the drug to reach the brain [48,49].’’

  • We made the change suggested for the reviewer to clarify the sentence.

In the Table 2 at pag 10 the authors introduced the PLLA  abbreviation which should be defined as poly(l-lactic acid)

  • We thank this observation. We have now defined the acronym.

In the Table 2 at reference [45] the abbreviation pDNA should be defined

  • We thank this observation. We have now defined the abbreviation.

Row 344  The phrase ‘’generally by hydrolysis (methanolysis) of polyvinyl-acetate (PVAc) “ should be clarified as ‘’generally by hydrolysis or methanol transesterification (methanolysis) of polyvinyl-acetate (PVAc) “

  • We made the change suggested for the reviewer to clarify the sentence.

Row 346  The claim regarding PVA  ‘’that is resistant to many solvents‘’ should be detailed with examples and clases of solvents

  • Thank you for this pertinent observation. We added more information and some examples.

Other acronyms that should be defined: row 402 CS-DS-Dox; rows 404, 405 PAA from PAA-b-PVA; row 407 HEC from PVA-HEC; row 416 PMMA from CS-g-PMMA:PVA-g-PMMA; also the meaning of letter g should be introduced (I presume that indicates the grafting of polymeric chains)

  • We have now defined these acronyms.

Row 429 The phrase "which could be beneficial or unfortunate. " should be corrected as "which could be beneficial or harmful. "

  • We made the change suggested for the reviewer to clarify the sentence.

Rows 472-475 The following claims " Interestingly, several reports demonstrated that poloxamers could decrease the cell membrane microviscosity (membrane fluidization) due to hydrophobic PPO chains. Furthermore, it has been reported that poloxamers could decrease multidrug resistance and induce a dramatic reduction in ATP levels in cancer and barrier cells and  inhibit drug efflux transporters. " should be supported by adequate references

  • According to this observation, we added two appropriate references.

Row 584  The chemical composition/name  of Brij35 (C12H25(OCH2CH2)23OH and SDS (Sodium dodecyl sulfate) should be introduced

  • We apologize for this omission. We added the suggested information.

Row 617 The abbreviation SE from Laurate SE should be defined

  • We thank this observation. We have now defined the acronym.

Rows 618-619 The abbreviation cremophor EL should be defined

  • We thank this observation. We have now defined the acronym.

Row 629    Abbreviation AUC should be defined

  • We thank this observation. We have now defined the acronym.

Row 638    Abbreviation ICH should be defined

  • We thank this observation. We have now defined the acronym.

A point which is must be introduced in this review is the immunogenic potential of these surfactants, especially those that contain polyethoxylated chains - see for example the review article [Nicola d’Avanzo, Christian Celia, Antonella Barone, Maria Carafa, Luisa Di Marzio, Hélder A. Santos, Massimo Fresta Immunogenicity of Polyethylene Glycol Based Nanomedicines: Mechanisms, Clinical Implications and Systematic Approach,  Adv. Therap. 2020, 1900170]

  • In concordance with this observation, we added information about the potential toxicity and immunogenicity of these compounds.

Reviewer 3 Report

This paper shows a good review of an analysis of the three principal non-ionic surfactants utilized in the manufacture of polymeric nanoparticles' performance. There are some issues that need to address:

- Introduction is written simply, most recent research and innovation in polysorbates and polymeric nanoparticle performances should be reviewed to show the gap of knowledge. The introduction should be extended with recent research papers.

- 277 References! Fantastic! Also, authors can cite the following work in the introduction which is closely related to their work and recently reported:

"Effect of UV and gamma irradiation sterilization processes in the properties of different polymeric nanoparticles for biomedical applications." Materials 13, no. 5 (2020): 1090.

"Evaluation of clay and fumed silica nanoparticles on adsorption of surfactant polymer during enhanced oil recovery." Journal of the Japan Petroleum Institute 60, no. 2 (2017): 85-94.

- what makes this review different from the other and from the most recent ones?

- section of drawbacks and future could be increased quality of the manuscript.

- There are some grammatical errors, please carefully check the whole manuscript.

- A review paper not only should summarize recently published works, but also should contain critical and comprehensive discussions. Therefore, check writing for the whole manuscript. The review should not be presented by listing what has done by others.

Congratulations to the authors for their professional work. This article can be very useful for researchers and students.

Author Response

Dear reviewer. Thank you for your valuable comments, your observations allowed us to strengthen our manuscript.

Reviewer 3

This paper shows a good review of an analysis of the three principal non-ionic surfactants utilized in the manufacture of polymeric nanoparticles' performance. There are some issues that need to address:

- Introduction is written simply, most recent research and innovation in polysorbates and polymeric nanoparticle performances should be reviewed to show the gap of knowledge. The introduction should be extended with recent research papers.

- Thank you, your valuable observation allowed us to visualize a new presentation of the introduction, and now we include a strengthened version.

- 277 References! Fantastic! Also, authors can cite the following work in the introduction which is closely related to their work and recently reported:

"Effect of UV and gamma irradiation sterilization processes in the properties of different polymeric nanoparticles for biomedical applications." Materials 13, no. 5 (2020): 1090.

"Evaluation of clay and fumed silica nanoparticles on adsorption of surfactant polymer during enhanced oil recovery." Journal of the Japan Petroleum Institute 60, no. 2 (2017): 85-94.

  • We highly appreciate this pertinent comment. We already carried out the necessary modifications.

What makes this review different from the other and from the most recent ones?

- We appreciate this valuable observation. This review differs from previous articles by focusing on the three stabilizers most used to prepare nanoparticles for biomedical applications. Of course, they are not the only ones, but they are the most frequently applied. With the premise of its wide use, we incorporated tables with many examples of nanoparticles for each stabilizer, offering the reader the possibility of a quick consultation. Furthermore, we offer structured sections to compare the properties of each stabilizer, ending with an advantages and disadvantages section. Last but not least, we included a general analysis of a systematic approach in the experimental procedure during the selection of a surfactant using the Quality by Design strategy.

- section of drawbacks and future could be increased quality of the manuscript.

- Thank you for this kind and pertinent suggestion. Now we include the new section from a critical and purposeful point of view.

- There are some grammatical errors, please carefully check the whole manuscript.

- We conducted a careful revision of our manuscript and corrected grammar errors.

- A review paper not only should summarize recently published works, but also should contain critical and comprehensive discussions. Therefore, check writing for the whole manuscript. The review should not be presented by listing what has done by others.

- We agree with this statement. With the valuable help of the reviewers' comments, we have made all the suggested changes that have allowed us to now offer a new version strengthened with critical aspects. Additionally, the "advantages and disadvantages" sections offer a critical perspective, while the "Quality by design in the choice of a surfactant" and "drawbacks and future" sections provide a comprehensive discussion of the manuscript.

Congratulations to the authors for their professional work. This article can be very useful for researchers and students.

  • We thank your helpful and kind observations.

Round 2

Reviewer 1 Report

I recommend it for publication in the present form.

Reviewer 2 Report

The authors improved the quality of their Review paper concerning three main  classes of polymeric  surfactants with applications in medicine according to my observations and suggestions

Reviewer 3 Report

The paper has been improved and corresponding modifications have been conducted. I think the current version can be considered for publication.